# CUPID: A Plug-in Framework for Joint Aleatoric and Epistemic Uncertainty Estimation with a Single Model

**Xinran Xu**[1]**, Xiuyi Fan**[1,2,3] *
[1]Lee Kong Chian School of Medicine, Nanyang Technological University, Singapore
[2]College of Computing and Data Science, Nanyang Technological University, Singapore
[3]Centre for Medical Technologies & Innovations, National Health Group, Singapore
`xinran007@e.ntu.edu.sg, xyfan@ntu.edu.sg`

## Abstract

Accurate estimation of uncertainty in deep learning is critical for deploying models in high-stakes domains such as medical diagnosis and autonomous decision-making, where overconfident predictions can lead to harmful outcomes. In practice, understanding the reason behind a model's uncertainty and the type of uncertainty it represents can support risk-aware decisions, enhance user trust, and guide additional data collection. However, many existing methods only address a single type of uncertainty or require modifications and retraining of the base model, making them difficult to adopt in real-world systems. We introduce CUPID (Comprehensive Uncertainty Plug-in estImation moDel), a general-purpose module that jointly estimates aleatoric and epistemic uncertainty without modifying or retraining the base model. CUPID can be flexibly inserted into any layer of a pretrained network. It models aleatoric uncertainty through a learned Bayesian identity mapping and captures epistemic uncertainty by analyzing the model's internal responses to structured perturbations. We evaluate CUPID across a range of tasks, including classification, regression, and out-of-distribution detection. The results show that it consistently delivers competitive performance while offering layer-wise insights into the origins of uncertainty. By making uncertainty estimation modular, interpretable, and model-agnostic, CUPID supports more transparent and trustworthy AI. Related code and data are available at https://github.com/a-Fomalhaut-a/CUPID.

## 1 Introduction

Deep neural networks have achieved impressive performance across many domains, yet they often lack reliable mechanisms for expressing uncertainty, leading to overconfident predictions and reduced trustworthiness (Li et al., 2023; Gawlikowski et al., 2023). Robust uncertainty estimation is essential for identifying misclassifications, detecting out-of-distribution inputs, and facilitating human involvement in decision making within safety critical environments (Yu et al., 2024).

Uncertainty in deep learning is generally divided into two types: aleatoric uncertainty, which arises from inherent noise or ambiguity in the data, and epistemic uncertainty, which reflects limitations in the model or training data (Der Kiureghian & Ditlevsen, 2009; Zou et al., 2023). Some studies further refine epistemic uncertainty into distributional uncertainty, caused by domain shifts, and model uncertainty, due to insufficient training or architectural constraints (Ulmer, 2021).

Numerous methods have been proposed to estimate uncertainty in deep learning models (Franchi et al., 2022; Zhang et al., 2024), but most focus on only one type or fail to clearly distinguish between aleatoric and epistemic components. This distinction is essential for decision-making in high-stakes domains like medical imaging (Hüllermeier & Waegeman, 2021). For instance, in diabetic retinopathy screening, high aleatoric uncertainty may signal poor image quality due to noise or blur,

---

*Corresponding author.

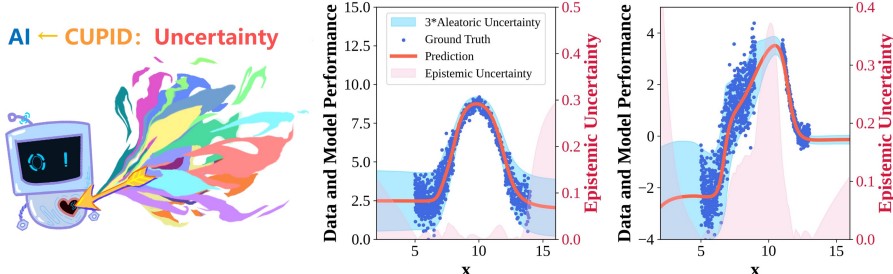

Figure 1: CUPID uncertainty estimation on a 1D regression toy problem. CUPID is inserted into an MLP-based predictive model. CUPID captures both aleatoric (blue) and epistemic (red) uncertainty.

while high epistemic uncertainty suggests the model is unfamiliar with certain pathological patterns. Disentangling these sources guides appropriate actions such as image reacquisition, expert review, or model refinement, ultimately improving system reliability. While some joint estimation methods exist, they often rely on specialized architectures such as Bayesian neural networks (Kendall & Gal, 2017) or diffusion models (Chan et al., 2024), and typically require retraining from scratch. This results in high computational cost and limits compatibility with existing systems.

In this work, we propose **CUPID** (Comprehensive Uncertainty Plug-in estImation moDel), a lightweight and versatile module that estimates both aleatoric and epistemic uncertainty with a single model, without requiring any alterations to model structure or retraining. Much like how Cupid's arrows unveil hidden affections, our CUPID model disentangles uncertainties within predictive models. Specifically, CUPID estimates aleatoric uncertainty by learning a Bayesian identity mapping while quantifying epistemic uncertainty by analyzing the model's internal responses under structured perturbations. Consider the simple 1D regression task shown in Figure 1. The model is trained on noisy samples with varying density and continuity. Based on the CUPID results, regions with high observation noise yield higher aleatoric uncertainty, while regions with little or no training coverage, such as edges and discontinuities, exhibit high epistemic uncertainty. This demonstrates the good uncertainty discouple performance of CUPID and the value of distinguishing between uncertainty types—not only for identifying prediction confidence, but also for understanding the underlying causes of model doubt.

By inserting CUPID at various intermediate layers, we are able to analyze how uncertainty evolves throughout the network, offering insight into where and how different types of uncertainty emerge during inference. Our experiments reveal that epistemic uncertainty tends to accumulate in the deeper parts of the network, where the model's representations become more abstract and task-specific. While integrating information from multiple layers can refine the estimates, the final layers are particularly informative for identifying epistemic uncertainty. In parallel, aleatoric uncertainty is more effectively captured from deeper feature representations, where variability in the input data is more prominently encoded. Beyond its simplicity, CUPID is broadly applicable to both classification and regression tasks. In summary, our key contributions are:

- Propose CUPID, a plug-in uncertainty estimation module capable of jointly estimating aleatoric and epistemic uncertainty without retraining the base model.

- Demonstrate CUPID's effectiveness across misclassification detection, out-of-distribution (OOD) detection, and regression tasks, achieving state-of-the-art performance on established uncertainty-aware benchmarks.

- Investigate how uncertainty evolves through network layers by inserting CUPID at different depths, offering a new perspective on the dynamics of uncertainty propagation.

## 2 RELATED WORKS

Uncertainty estimation plays a critical role in enhancing the reliability, safety, and interpretability of deep learning systems (Abdar et al., 2021; Liang et al., 2022). Broadly, existing methods can be grouped into two categories based on whether they require modifications to the predictive model's

parameters: model-preserving approaches, which estimate uncertainty without altering or retraining the base model, and model-redefining approaches, which involve architectural changes or full retraining to capture uncertainty within a new probabilistic framework.

**Model-redefining approaches**   These methods require modifying or retraining the predictive model to integrate uncertainty estimation. Bayesian Neural Networks (BNNs) treat model weights as distributions, capturing both aleatoric and epistemic uncertainty, but are computationally intensive due to the need for retraining and posterior sampling (Blundell et al., 2015; Kendall & Gal, 2017; Maddox et al., 2019). Evidential Deep Learning (EDL) models predictive distributions via a Dirichlet framework, interpreting output logits as evidence and distinguishes uncertainty types using distributional properties (Sensoy et al., 2018; Ye et al., 2024). Deep ensembles aggregate predictions from multiple independently trained models to estimate uncertainty through predictive variance (Lakshminarayanan et al., 2017; Durasov et al., 2021; Wen et al., 2020). While effective, these approaches incur high training overhead and are less practical for large-scale applications.

To address these challenges, HyperDM (Chan et al., 2024) integrates Bayesian hyper-networks with conditional diffusion models. It approximates the benefits of deep ensembles at a fraction of the computational cost. HyperDM highlights the potential of model-redefining approaches for extending uncertainty estimation to complex, high-dimensional problems, though its reliance on diffusion architectures may limit applicability where other model families are preferred.

**Model-preserving approaches**   These methods estimate uncertainty without altering the original model architecture. Test-time augmentation strategies estimate uncertainty by measuring prediction variability across transformed inputs (input rotation or noise perturbation) (Wang et al., 2018a; Mi et al., 2022). MC Dropout applies dropout at inference to approximate Bayesian sampling, but suffers from increased inference time due to multiple forward passes (Gal & Ghahramani, 2016; Leibig et al., 2017). Gradient-based methods have proven effective in approximating epistemic uncertainty by using gradient norms as a proxy (Riedlinger et al., 2023; Wang & Ji, 2024). More recently, uncertainty has also been estimated by designing auxiliary loss functions that enable gradient computation without requiring ground truth labels (Hornauer et al., 2025).

Alternatively, training an additional model offers a practical solution. BayesCap (Upadhyay et al., 2022) learns to estimate uncertainty on top of frozen pre-trained outputs, enabling efficient uncertainty quantification. Rate-In Zeevi et al. (2025) extends MC Dropout by adding dropout layers and treating dropout as a tunable component at inference time. By quantifying information loss in feature maps, it adaptively adjusts dropout rates per layer and per input, making dropout behave like a trainable model rather than a fixed regularizer. RUE (Wang et al., 2023) estimates distributional shift via reconstruction error, while other works (Yu et al., 2024) explicitly separate aleatoric and epistemic uncertainty estimation with two dedicated modules. These approaches balance efficiency and flexibility, making them suitable for deployment in real-world settings.

## 3   METHOD

### 3.1   PROBLEM FORMULATION

We consider a supervised learning setting in which a neural network model $M : \mathbb{X} \to \mathbb{Y}$ is trained to map input data $\mathbf{x} \in \mathbb{X}$ to corresponding targets $\mathbf{y} \in \mathbb{Y}$. The training dataset is defined as a finite set of $N$ labeled examples:

$$\mathcal{D} = \{(\mathbf{x}_n, \mathbf{y}_n)\}_{n=1}^N \subseteq \mathbb{X} \times \mathbb{Y}. \tag{1}$$

Given a new input sample $\mathbf{x}^* \in \mathbb{X}$, the predictive model $M$, parameterized by weights $\boldsymbol{\theta}$, produces a prediction $\hat{\mathbf{y}}^* = M(\mathbf{x}^*; \boldsymbol{\theta})$. In a Bayesian formulation, the predictive distribution over the target output is given by marginalizing over the posterior distribution of model parameters:

$$p(\mathbf{y}^* \mid \mathbf{x}^*, \mathcal{D}) = \int \underbrace{p(\mathbf{y}^* \mid \mathbf{x}^*, \boldsymbol{\theta})}_{\text{Aleatoric}} \underbrace{p(\boldsymbol{\theta} \mid \mathcal{D})}_{\text{Epistemic}} d\boldsymbol{\theta}. \tag{2}$$

This decomposition reveals two sources of uncertainty: aleatoric uncertainty, which arises from inherent noise in the data, and epistemic uncertainty, which reflects the model's uncertainty about its own parameters. Our goal is to estimate both types of uncertainty using a unified framework.

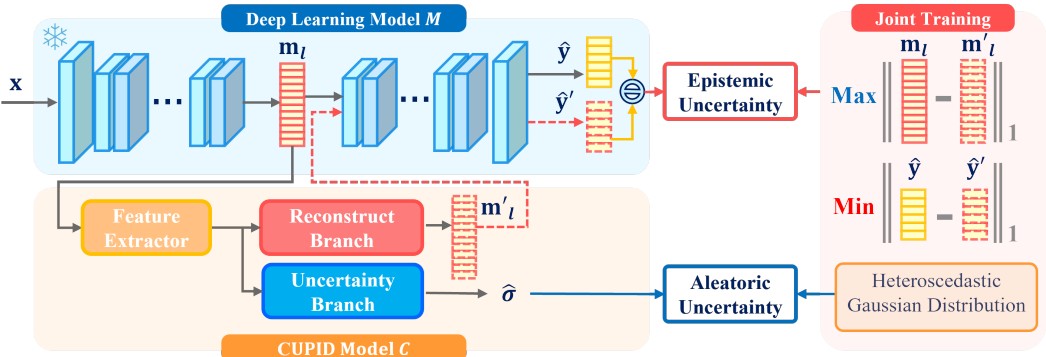

Figure 2: The CUPID pipeline. Aleatoric uncertainty is estimated using a dedicated Uncertainty Branch, while epistemic uncertainty is captured by measuring the variance between the original model output $\hat{\mathbf{y}}$ and the perturbed output $\hat{\mathbf{y}}'$.

To this end, we introduce CUPID, a plug-in uncertainty estimation module that can be flexibly inserted at any intermediate layer $l$ of the predictive model $M$. CUPID consists of three main components: a *Feature Extractor*, a *Reconstruction Branch*, and an *Uncertainty Branch*. It operates on the intermediate feature representation at a selected layer and outputs both a perturbed feature and uncertainty estimates. Formally, consider a predictive model $M$ decomposed as a composition of two sub-networks:

$$M(\mathbf{x}) = F_l(B_l(\mathbf{x})), \tag{3}$$

where $B_l : \mathbb{X} \to \mathbb{R}^d$ extracts the intermediate feature $\mathbf{m}_{l,n} = B_l(\mathbf{x}_n)$ at layer $l$ with $d$ dimension, and $F_l : \mathbb{R}^d \to \mathbb{Y}$ maps it to the final prediction.

The CUPID module $C : \mathbb{R}^d \to \mathbb{R}^d \times \mathbb{R}^k$, parameterized by $\boldsymbol{\omega}$, operates on $\mathbf{m}_{l,n}$ and outputs a reconstructed feature $\mathbf{m}'_{l,n} \in \mathbb{R}^d$ and an aleatoric uncertainty estimate $\hat{\boldsymbol{\sigma}}_n \in \mathbb{R}^k$:

$$(\mathbf{m}'_{l,n}, \hat{\boldsymbol{\sigma}}_n) = C(\mathbf{m}_{l,n}; \boldsymbol{\omega}). \tag{4}$$

The reconstructed feature $\mathbf{m}'_{l,n}$ is forwarded through the remainder of the network to produce a perturbed prediction:

$$\hat{\mathbf{y}}' = F_l(\mathbf{m}'_{l,n}). \tag{5}$$

Epistemic uncertainty is quantified as the discrepancy between the original prediction $\hat{\mathbf{y}}_n = F_l(\mathbf{m}_{l,n})$ and the perturbed prediction $\hat{\mathbf{y}}'_n$:

$$U_{\text{epis}}(\mathbf{x}) := \|\hat{\mathbf{y}}_n - \hat{\mathbf{y}}'_n\|_1. \tag{6}$$

By explicitly modeling both the reconstruction of features and predictive variation, CUPID enables interpretable estimation of both aleatoric and epistemic uncertainties at any specified internal layer of the model.

## 3.2 ALEATORIC UNCERTAINTY ESTIMATION WITH CUPID

Aleatoric uncertainty refers to the inherent noise present in the data, arising from factors such as measurement error, sensor limitations, or ambiguous inputs. This type of uncertainty is irreducible and persists even with unlimited training data. A common strategy to model aleatoric uncertainty is to assume that the network's output is corrupted by observation noise, which follows a heteroscedastic Gaussian distribution with input-dependent variance (Upadhyay et al., 2022).

Specifically, for each input $\mathbf{x}_n$, the predictive distribution over the target $\mathbf{y}_n$ is modeled as:

$$p(\mathbf{y}_n \mid \mathbf{x}_n, \boldsymbol{\theta}, \boldsymbol{\omega}) = \mathcal{N}(\hat{\mathbf{y}}'_n, \hat{\boldsymbol{\sigma}}_n^2), \tag{7}$$

where $\hat{\boldsymbol{\sigma}}_n^2 \in \mathbb{R}^k$ is the predicted data-dependent variance output by the Uncertainty Branch of CUPID. $k$ equals the output dimension.

Under this probabilistic modeling assumption, the optimal parameters of the Uncertainty Branch, $\boldsymbol{\omega}$, are obtained by maximizing the log-likelihood over the dataset:

$$\boldsymbol{\omega}^* = \arg\max_{\boldsymbol{\omega}} \sum_{n=1}^{N} \log p(\mathbf{y}_n \mid \mathbf{x}_n, \boldsymbol{\theta}, \boldsymbol{\omega})$$

$$= \arg\max_{\boldsymbol{\omega}} \sum_{n=1}^{N} \left[ -\frac{\|\hat{\mathbf{y}}'_n - \mathbf{y}_n\|_2^2}{2\hat{\boldsymbol{\sigma}}_n^2} - \frac{1}{2}\log(\hat{\boldsymbol{\sigma}}_n^2) \right]. \tag{8}$$

The predicted variance $\hat{\boldsymbol{\sigma}}_n^2$ then serves as an estimate of the aleatoric uncertainty for sample $n$:

$$U_{\text{alea}}(\mathbf{x}_n) := \hat{\boldsymbol{\sigma}}_n^2. \tag{9}$$

To improve numerical stability during optimization, we follow the standard approach of predicting the log-variance $\mathbf{s}_n = \log(\hat{\boldsymbol{\sigma}}_n^2)$ rather than the variance itself (Kendall & Gal, 2017). The resulting loss function for the Uncertainty Branch becomes:

$$\mathcal{L}_{\text{alea}} = \frac{1}{N} \sum_{n=1}^{N} \left[ \frac{1}{2}\exp(-\mathbf{s}_n)\|\mathbf{y}_n - \hat{\mathbf{y}}'_n\|_2^2 + \frac{1}{2}\mathbf{s}_n \right]. \tag{10}$$

While the formulation above is presented in a regression setting, the same likelihood principle extends naturally to classification. In this case, the model produces logits $\hat{\mathbf{z}}'_n$, which represent unnormalized evidence for each class; applying the Softmax yields the predictive probability vector $\hat{\mathbf{y}}'_n = \text{Softmax}(\hat{\mathbf{z}}'_n)$. Both $\hat{\mathbf{y}}'_n$ and the one-hot label $\mathbf{y}_n$ can be viewed as continuous distributions. This allows defining a Brier-style heteroscedastic objective over $\|\mathbf{y}_n - \hat{\mathbf{y}}'_n\|_2^2$.

### 3.3 EPISTEMIC UNCERTAINTY ESTIMATION WITH CUPID

Epistemic uncertainty captures the model's lack of knowledge, often attributed to limited training data or uncertainty in model parameters. This type of uncertainty can be reduced with more data and typically increases in regions of the input space that are underrepresented during training. Additionally, epistemic uncertainty is closely associated with distributional shifts, arising when test samples deviate from the training distribution. CUPID estimates epistemic uncertainty by encouraging the Reconstruction Branch to produce a feature perturbation that is maximally different from the original intermediate feature $\mathbf{m}_{l,n}$, while maintaining the same output prediction. Formally, we seek to find a reconstructed feature $\mathbf{m}'_{l,n}$ that satisfies:

$$\underset{\mathbf{m}'_{l,n}}{\text{maximize}} \ \|\mathbf{m}'_{l,n} - \mathbf{m}_{l,n}\|_1 \quad \text{and} \quad \underset{\mathbf{m}'_{l,n}}{\text{minimize}} \ \|\hat{\mathbf{y}}'_n - \hat{\mathbf{y}}_n\|_1. \tag{11}$$

The loss function to train the Reconstruction Branch therefore balances a differential feature term that promotes large deviations with a prediction consistency constraint:

$$\mathcal{L}_{\text{epis}} = \frac{1}{N} \sum_{n=1}^{N} \left[ \|\hat{\mathbf{y}}_n - \hat{\mathbf{y}}'_n\|_1 - \lambda_1 \|\mathbf{m}'_{l,n} - \mathbf{m}_{l,n}\|_1 \right], \tag{12}$$

where $\lambda_1 > 0$ is a hyperparameter that controls the trade-off between prediction invariance and feature perturbation magnitude. To avoid trivial solutions where the perturbation grows arbitrarily, we initialize CUPID close to the identity mapping. The epistemic uncertainty is then quantified by:

$$U_{\text{epis}}(\mathbf{x}) := \|F_l(\mathbf{m}_{l,n}) - F_l(\mathbf{m}'_{l,n})\|_1. \tag{13}$$

To further interpret this measure, we consider a first-order Taylor expansion of $F_l$ around $\mathbf{m}_{l,n}$, assuming local differentiability, then we obtain the approximation:

$$U_{\text{epis}}(\mathbf{x}) \approx \|\nabla_{\mathbf{m}_{l,n}} F_l(\mathbf{m}_{l,n}) \cdot (\mathbf{m}'_{l,n} - \mathbf{m}_{l,n})\|_1. \tag{14}$$

This formulation reveals two key components that jointly determine the magnitude of epistemic uncertainty:

$$U_{\text{epis}}(\mathbf{x}) \propto \text{Sensitivity} \times \text{Deviation}, \tag{15}$$

where the Jacobian $\nabla_{\mathbf{m}_{l,n}} F_l(\mathbf{m}_{l,n})$ reflects the local sensitivity of the model's output to perturbations in feature space. The perturbation $\|\mathbf{m}'_{l,n} - \mathbf{m}_{l,n}\|_1$ captures the extent to which the input

deviates from the training manifold. In-distribution misclassified samples often exhibit high sensitivity, while OOD samples induce abnormally large deviation. CUPID therefore provides a unified estimate of epistemic uncertainty that responds to both failure modes. For classification tasks where softmax activation is used to produce probability distributions over discrete classes, the output discrepancy is computed in the softmax space.

### 3.4 LOSS FUNCTION

To jointly estimate epistemic and aleatoric uncertainty, the total loss is defined as:

$$\mathcal{L}_{\text{CUPID}} = \mathcal{L}_{\text{epis}} + \lambda_2\,\mathcal{L}_{\text{alea}}, \tag{16}$$

where $\lambda_2$ is a weighting hyperparameter balancing the aleatoric loss $\mathcal{L}_{\text{alea}}$ against the epistemic loss $\mathcal{L}_{\text{epis}}$. Both the epistemic and aleatoric estimation are optimized simultaneously under this unified loss, ensuring that CUPID learns both uncertainty types within a single model.

## 4 EXPERIMENTS

In this section, we systematically evaluate CUPID's effectiveness in estimating both aleatoric and epistemic uncertainty across three distinct tasks: medical image misclassification detection, out-of-distribution detection, and image super-resolution. These tasks are selected to highlight the generalizability of CUPID across classification and regression problems, as well as across high-stakes and general-purpose domains. We also perform an ablation study to assess the impact of placing CUPID at different locations within the model architecture and to analyze the influence of internal hyperparameters on its performance. Each experiment was repeated three times, and we report the mean and standard deviation for all evaluation metrics. The detailed model architectures, implementation specifics, and main task performance metrics for all experiments are provided in the appendix.

### 4.1 MEDICAL IMAGE MISCLASSIFICATION DETECTION

Table 1: Performance of misclassification detection (misclassified samples as positive). The best model for each metric is in bold, and the second best is underlined. CUPID Aleatoric achieved the best performance on GLV2, while CUPID Epistemic performed best on HAM10000, suggesting different dominant sources of uncertainty across datasets.

| Method | GLV2 | | | HAM10000 | | |
|---|---|---|---|---|---|---|
| | AUC (↑) | AURC (↓) | Spearman (↑) | AUC (↑) | AURC (↓) | Spearman (↑) |
| CUPID Alea. | **0.870 ± 0.002** | **0.018 ± 0.001** | 0.941 ± 0.004 | 0.769 ± 0.023 | 0.067 ± 0.007 | 0.722 ± 0.014 |
| CUPID Epis. | 0.769 ± 0.015 | 0.034 ± 0.002 | 0.701 ± 0.051 | **0.855 ± 0.006** | **0.047 ± 0.001** | 0.907 ± 0.001 |
| MC Dropout | 0.768 ± 0.006 | 0.027 ± 0.001 | 0.888 ± 0.005 | 0.829 ± 0.001 | 0.076 ± 0.001 | 0.861 ± 0.002 |
| Rate-in | 0.815 ± 0.006 | 0.024 ± 0.001 | 0.816 ± 0.004 | 0.846 ± 0.001 | 0.048 ± 0.000 | **0.915 ± 0.000** |
| IGRUE | 0.642 ± 0.007 | 0.058 ± 0.002 | 0.199 ± 0.004 | 0.548 ± 0.004 | 0.157 ± 0.002 | 0.027 ± 0.018 |
| PostNet Alea. | 0.671 ± 0.006 | 0.182 ± 0.004 | 0.641 ± 0.011 | 0.793 ± 0.007 | 0.142 ± 0.003 | 0.764 ± 0.006 |
| PostNet Epis. | 0.559 ± 0.031 | 0.238 ± 0.019 | 0.284 ± 0.054 | 0.751 ± 0.017 | 0.158 ± 0.010 | 0.698 ± 0.033 |
| BNN | 0.829 ± 0.018 | 0.025 ± 0.003 | **0.954 ± 0.007** | 0.793 ± 0.006 | 0.096 ± 0.004 | 0.821 ± 0.009 |
| DEC | 0.503 ± 0.012 | 0.192 ± 0.006 | 0.803 ± 0.139 | 0.837 ± 0.017 | 0.082 ± 0.004 | 0.874 ± 0.007 |

**Experiment setting** We begin our evaluation with medical image classification, a domain where reliability, interpretability, and calibrated uncertainty are critical for deployment. In clinical settings, a model's ability to identify its mispredictions can directly impact diagnostic decisions and patient safety. To this end, we evaluate CUPID on misclassification detection using two medical imaging benchmarks: GLV2 (glaucoma detection) (Gulshan et al., 2016; Kiefer et al., 2022) and HAM10000 (skin lesion classification) (Tschandl et al., 2018).

Baselines include Rate-in (Zeevi et al., 2025), MC Dropout (Folgoc et al., 2021), PostNet (Charpentier et al., 2020), IGRUE (Korte et al., 2024), DEC (Sensoy et al., 2018), and BNN (Kendall & Gal, 2017), all implemented with ResNet18 (He et al., 2016) for consistency. CUPID is integrated after the final residual block of ResNet18. We report AUC, AURC (Ding et al., 2020), and Spearman's rank correlation (Rasmussen et al., 2023). AUC measures the ability to separate correct from

incorrect predictions for misclassification detection; AURC assesses the confidence-error trade-off; and Spearman quantifies the correlation between uncertainty and error.

**Results** The results of misclassification detection on the GLV2 and HAM10000 datasets are presented in Table 1. CUPID Aleatoric achieves the highest AUC (0.870) and lowest AURC (0.018) on GLV2. Its Spearman score (0.941) is also competitive with BNN (0.954). These results suggest that data-driven noise is the dominant uncertainty source in the GLV2-trained model.

On HAM10000, CUPID Epistemic achieves the best performance (AUC: 0.855, AURC: 0.047, Spearman: 0.907), highlighting the significance of model-based uncertainty in more diverse skin lesion data. Rate-in, DEC and MC Dropout also perform better on HAM1000, reflecting their sensitivity to epistemic uncertainty. These results confirm CUPID's ability to disentangle and capture different uncertainty sources, with the dominant type varying by dataset. This underscores the need for clear uncertainty modeling in domain-specific applications.

## 4.2 OOD DETECTION

Table 2: Performance of OOD detection (OOD samples as positive). PAPILA and ACRIMA share the same research problem (glaucoma detection) with the ID dataset while CIFAR10 is a general classification dataset.

| Method | PAPILA | | ACRIMA | | CIFAR10 | |
|---|---|---|---|---|---|---|
| | AUC($\uparrow$) | AUPR($\uparrow$) | AUC($\uparrow$) | AUPR($\uparrow$) | AUC($\uparrow$) | AUPR($\uparrow$) |
| CUPID Alea. | $0.379 \pm 0.027$ | $0.333 \pm 0.007$ | $0.717 \pm 0.029$ | $0.661 \pm 0.027$ | $\mathbf{0.983 \pm 0.005}$ | $\mathbf{0.998 \pm 0.001}$ |
| CUPID Epis. | $\mathbf{0.877 \pm 0.032}$ | $\mathbf{0.854 \pm 0.027}$ | $\mathbf{0.978 \pm 0.010}$ | $\mathbf{0.984 \pm 0.007}$ | $0.898 \pm 0.054$ | $\underline{0.991 \pm 0.005}$ |
| MC Dropout | $\underline{0.733 \pm 0.002}$ | $0.586 \pm 0.007$ | $0.869 \pm 0.003$ | $0.816 \pm 0.009$ | $0.887 \pm 0.004$ | $0.986 \pm 0.001$ |
| Rate-in | $0.328 \pm 0.005$ | $0.329 \pm 0.008$ | $0.363 \pm 0.003$ | $0.390 \pm 0.003$ | $0.620 \pm 0.001$ | $0.927 \pm 0.002$ |
| IGRUE | $0.636 \pm 0.114$ | $0.486 \pm 0.097$ | $\underline{0.941 \pm 0.008}$ | $\underline{0.944 \pm 0.008}$ | $\underline{0.978 \pm 0.005}$ | $\mathbf{0.998 \pm 0.001}$ |
| PostNet Alea. | $0.638 \pm 0.060$ | $0.487 \pm 0.067$ | $0.549 \pm 0.040$ | $0.487 \pm 0.040$ | $0.657 \pm 0.032$ | $0.952 \pm 0.005$ |
| PostNet Epis. | $0.577 \pm 0.097$ | $0.425 \pm 0.088$ | $0.685 \pm 0.154$ | $0.654 \pm 0.151$ | $0.773 \pm 0.082$ | $0.976 \pm 0.011$ |
| BNN | $0.707 \pm 0.040$ | $\underline{0.612 \pm 0.050}$ | $0.708 \pm 0.073$ | $0.699 \pm 0.042$ | $0.643 \pm 0.108$ | $0.959 \pm 0.013$ |
| DEC | $0.515 \pm 0.024$ | $0.457 \pm 0.024$ | $0.680 \pm 0.003$ | $0.685 \pm 0.012$ | $0.660 \pm 0.015$ | $0.963 \pm 0.003$ |

**Experiment setting** We evaluate OOD detection using GLV2 as the in-distribution (ID) dataset with ACRIMA (Diaz-Pinto et al., 2019), PAPILA (Kovalyk et al., 2022), and CIFAR-10 (Krizhevsky & Hinton, 2009) as out-of-distribution (OOD) datasets. ACRIMA and PAPILA, though also related to glaucoma detection, differ in image quality and focus: PAPILA has lower contrast and reddish tones, while ACRIMA highlights optic disc regions through cropping. CIFAR-10 serves as a general OOD benchmark due to domain dissimilarity. Baselines follow those in misclassification detection. AUC and AUPR are used to measure performance, treating OOD samples as positive (Techapanurak & Okatani, 2021). AUPR highlights robustness under class imbalance.

**Results** The results are summarized in Table 2. Our proposed CUPID model demonstrates strong OOD detection performance across all datasets. CUPID Epistemic achieves the best AUPR and AUC on ACRIMA and PAPILA, highlighting its sensitivity to subtle distribution shifts within the same clinical task. Interestingly, CUPID Aleatoric performs best on CIFAR-10 (AUC 0.983, AUPR 0.998). CUPID Aleatoric models input-dependent (heteroscedastic) uncertainty through a learned variance term. It assigns high uncertainty when the input lies in feature space regions that are both underrepresented and unpredictable. This enables CUPID Aleatoric to respond robustly to extreme domain mismatches, explaining its superior performance on CIFAR-10.

Among baselines, IGRUE perform well on CIFAR-10 and ACRIMA but struggle with PAPILA. Rate-in and MC Dropout, despite strong misclassification detection results, underperform in OOD detection, likely due to overconfidence. Overall, CUPID adapts effectively to both in-task and cross-task shifts, with aleatoric and epistemic branches complementing each other across OOD types.

### 4.3 IMAGE SUPER-RESOLUTION AS REGRESSION TASK

**Experiment setting** We evaluate uncertainty estimation in super-resolution (SR) using a pre-trained ESRGAN model (Wang et al., 2018b) trained on DIV2K (Agustsson & Timofte, 2017). CUPID is integrated before the upsampling module of ESRGAN. For testing, we utilize three standard benchmarks: Set5 (Bevilacqua et al., 2012), Set14 (Zeyde et al., 2010), and BSDS100 (Martin et al., 2001). To assess generalization across modalities, we additionally include the IXI dataset (Biomedical Image Analysis Group, Imperial College London, 2022), a brain MRI dataset that differs substantially in appearance and domain. Specifically, we use T1-weighted MRI scans, which are grayscale and structurally different from the natural images in DIV2K. We compare CUPID with five baselines: BayesCap (Upadhyay et al., 2022), which reconstructs output distributions and learns a Bayesian identity mapping; in-rotate and in-noise, which measure output variation from input perturbations (Mi et al., 2022; Wang et al., 2019); and med-noise and med-dropout (Mi et al., 2022), which inject randomness into intermediate features.

Table 3: Performance on natural image datasets (Set5, Set14, BSDS100) and medical imaging dataset IXI (MRI scans). CUPID Aleatoric achieves the best results on the natural image benchmarks, while CUPID Epistemic performs best on the IXI dataset.

| Method | Set5 | | | Set14 | | |
| --- | --- | --- | --- | --- | --- | --- |
| | Pearson (↑) | AUSE (↓) | UCE (↓) | Pearson (↑) | AUSE (↓) | UCE (↓) |
| CUPID Alea. | **0.528 ± 0.006** | **0.010 ± 0.000** | **0.045 ± 0.018** | **0.527 ± 0.002** | **0.012 ± 0.000** | **0.049 ± 0.005** |
| CUPID Epis. | 0.416 ± 0.004 | 0.018 ± 0.001 | 0.266 ± 0.007 | 0.449 ± 0.005 | 0.019 ± 0.000 | 0.226 ± 0.003 |
| BayesCap | 0.485 ± 0.038 | **0.010 ± 0.000** | 0.098 ± 0.001 | 0.422 ± 0.064 | **0.012 ± 0.000** | 0.100 ± 0.000 |
| in-rotate | 0.493 ± 0.000 | **0.010 ± 0.000** | 0.071 ± 0.000 | 0.490 ± 0.000 | 0.013 ± 0.000 | 0.072 ± 0.000 |
| in-noise | 0.370 ± 0.006 | 0.019 ± 0.000 | 0.051 ± 0.035 | 0.354 ± 0.001 | 0.022 ± 0.000 | 0.826 ± 0.006 |
| med-dropout | 0.219 ± 0.023 | 0.030 ± 0.001 | 0.680 ± 0.043 | 0.271 ± 0.012 | 0.024 ± 0.000 | 0.292 ± 0.022 |
| med-noise | 0.312 ± 0.003 | 0.022 ± 0.000 | 0.826 ± 0.006 | 0.293 ± 0.002 | 0.022 ± 0.000 | 0.826 ± 0.006 |
| Method | BSDS100 | | | IXI | | |
| | Pearson (↑) | AUSE (↓) | UCE (↓) | Pearson (↑) | AUSE (↓) | UCE (↓) |
| CUPID Alea. | **0.536 ± 0.001** | 0.012 ± 0.000 | **0.042 ± 0.012** | 0.677 ± 0.008 | **0.004 ± 0.000** | **0.021 ± 0.004** |
| CUPID Epis. | 0.464 ± 0.007 | 0.018 ± 0.000 | 0.185 ± 0.007 | **0.734 ± 0.018** | **0.004 ± 0.000** | 0.298 ± 0.013 |
| BayesCap | 0.427 ± 0.034 | **0.011 ± 0.000** | 0.100 ± 0.000 | 0.447 ± 0.034 | **0.004 ± 0.000** | 0.100 ± 0.000 |
| in-rotate | 0.465 ± 0.000 | 0.012 ± 0.000 | 0.077 ± 0.000 | 0.598 ± 0.000 | **0.004 ± 0.000** | 0.093 ± 0.000 |
| in-noise | 0.353 ± 0.001 | 0.022 ± 0.000 | 0.826 ± 0.006 | 0.461 ± 0.001 | 0.005 ± 0.000 | 0.091 ± 0.002 |
| med-dropout | 0.397 ± 0.002 | 0.020 ± 0.000 | 0.136 ± 0.008 | 0.570 ± 0.001 | 0.007 ± 0.000 | 0.337 ± 0.026 |
| med-noise | 0.293 ± 0.000 | 0.024 ± 0.000 | 0.700 ± 0.002 | 0.439 ± 0.000 | 0.006 ± 0.000 | 0.859 ± 0.002 |

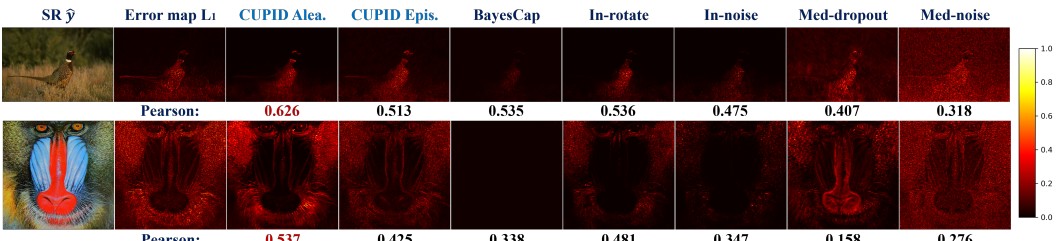

Figure 3: Comparison of visual results between error and uncertainty maps. CUPID Aleatoric shows the best texture alignment and highest correlation with error maps.

To evaluate the quality of uncertainty estimation in the regression problem, we adopt three complementary metrics. Pearson's correlation coefficient measures the linear relationship between predicted uncertainty and error. The Area Under the Sparsification Error Curve (AUSE) (Ilg et al., 2018) quantifies how well uncertainty identifies inaccurate predictions by evaluating deviation from an ideal sparsification curve. Finally, Uncertainty Calibration Error (UCE) (Laves et al., 2020) assesses alignment between predicted uncertainty and error across confidence intervals, reflecting how well the estimates are calibrated. The $L_1$ loss map is used as error to compute these metrics.

**Results**   Table 3 reports quantitative results. CUPID Aleatoric achieves superior performance across all natural image datasets (Pearson > 0.52, AUSE < 0.13, UCE < 0.05). BayesCap and in-rotate show moderate performance, while CUPID Epistemic and med-dropout perform poorly on the first three datasets. Methods relying on noise injection degrade performance, likely due to perturbation sensitivity. These results suggest aleatoric uncertainty is the dominant contributor to overall uncertainty in super-resolution tasks, rather than model uncertainty. On the IXI dataset, which differs significantly from the training distribution, CUPID Epistemic outperforms its aleatoric counterpart on Pearson, showing that epistemic uncertainty becomes more informative under domain shifts and highlighting CUPID's capacity to adapt to unfamiliar distributions. Figure 3 provides a visual comparison of uncertainty maps generated by different methods. CUPID's maps exhibit clearer structure and better alignment with actual error regions, reinforcing its advantage in uncertainty estimation.

### 4.4 HYPERPARAMETER EXPERIMENTS

**CUPID location**   To investigate how uncertainty evolves and originates during forward propagation, we conducted experiments by inserting CUPID at different intermediate layers of the predictive model, as summarized in Figure 4.

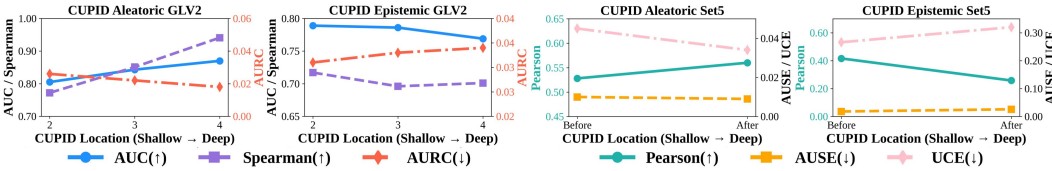

Figure 4: Performance of CUPID inserted at varying locations: misclassification detection (Left) and super-resolution (Right). Aleatoric uncertainty estimation improves when CUPID is placed closer to the output, while epistemic uncertainty benefits from earlier insertion points.

For the medical image classification task, CUPID was integrated after the 2nd, 3rd, and 4th stages of residual blocks in the ResNet-18 model. The results demonstrate a clear trend: aleatoric uncertainty is more accurately estimated when CUPID is placed closer to the output layer, while epistemic uncertainty benefits from earlier placements within the network. This observation aligns with the conceptual distinction between the two types of uncertainty. Aleatoric uncertainty, which originates from inherent noise in the input data, tends to manifest more prominently in high-level features near the prediction layer, where semantic decisions are made. The results show that estimating aleatoric uncertainty directly from input features is insufficient, whereas using deeper activations, especially those near the output, provides a more reliable signal. Conversely, epistemic uncertainty reflects the model's internal representation and parameter uncertainty. Placing CUPID in earlier layers enables it to better observe how uncertain representations propagate and interact throughout the model's depth. Notably, the strong epistemic performance observed with CUPID positioned near the output highlights that model uncertainty predominantly accumulates in the final layers. A similar trend is observed in the super-resolution setting. Specifically, when CUPID is inserted before (B) and after (A) the upsampling module in the ESRGAN, we observe that epistemic uncertainty is better captured in earlier layers, while aleatoric uncertainty estimation improves post-upsampling.

Table 4: Performance of differential feature loss on OOD task. "No max" means remove $-\|\mathbf{m}_{l,n} - \mathbf{m}'_{l,n}\|_1$ in the loss function. Best-performing results for each metric are highlighted in bold.

| Method | | PAPILA | | ACRIMA | | CIFAR10 | |
|---|---|---|---|---|---|---|---|
| | | AUC(↑) | AUPR(↑) | AUC(↑) | AUPR(↑) | AUC(↑) | AUPR(↑) |
| Max | Alea. | 0.379 ± 0.027 | 0.333 ± 0.007 | 0.717 ± 0.029 | 0.661 ± 0.027 | 0.983 ± 0.005 | 0.998 ± 0.001 |
| No max | Alea. | 0.389 ± 0.026 | 0.338 ± 0.009 | 0.739 ± 0.042 | 0.696 ± 0.055 | **0.988 ± 0.003** | **0.999 ± 0.000** |
| Max | Epis. | **0.877 ± 0.032** | **0.854 ± 0.027** | **0.978 ± 0.010** | **0.984 ± 0.007** | 0.898 ± 0.054 | 0.991 ± 0.005 |
| No max | Epis. | 0.839 ± 0.017 | 0.790 ± 0.054 | 0.977 ± 0.006 | 0.982 ± 0.005 | 0.875 ± 0.024 | 0.989 ± 0.002 |

**Loss function**   We further conduct an ablation study on the loss function. On the HAM10000 dataset, incorporating the differential feature loss ($-\|\mathbf{m}_{l,n} - \mathbf{m}'_{l,n}\|_1$) yields a slight improvement in aleatoric uncertainty estimation (Spearman ↑ 0.004) while maintaining comparable performance

for epistemic uncertainty. The benefits of this loss term become more evident in the OOD detection as shown in Table 4. On the PAPILA dataset, the addition of the differential feature loss leads to a substantial improvement in CUPID Epistemic's performance (AUC: 0.839-0.877, AUPR: 0.790-0.854). These results suggest that the differential feature loss enhances CUPID's sensitivity to distributional shifts and improves its ability under OOD conditions. Details and further studies are provided in the appendix.

**Joint vs. Separate Training.** Both the reconstruction branch (epistemic) and the uncertainty branch (aleatoric) in CUPID are present and jointly optimized. To evaluate whether this two-branch architecture provides mutual benefit, we conduct an ablation study in which we remove one branch entirely and train the remaining branch in isolation. Specifically, (1) *Alea. separate* denotes a model where the epistemic branch is removed and only the aleatoric branch is trained, and (2) *Epis. separate* denotes a model where the aleatoric branch is removed and only the epistemic branch is trained.

On the GLV2 misclassification detection task (Table 5), the fully joint model outperforms both single-branch variants across all metrics, indicating that each type of uncertainty estimation benefits from the presence of the other branch during training. In OOD detection (Table 6), the epistemic uncertainty from the joint model also achieves substantially higher AUC and AUPR than the epistemic-only variant (PAPILA AUC: 0.877-0.771), demonstrating that the joint formulation yields a more distribution-aware and robust representation.

This improvement arises from the complementary objectives of the two branches. For aleatoric uncertainty, the prediction-consistency constraint used in the epistemic loss, $\min_{\mathbf{m}'_{l,n}} \|\hat{\mathbf{y}}'_n - \hat{\mathbf{y}}_n\|_1$, regularizes the shared feature extractor by discouraging perturbation-sensitive or unstable representations. This yields better-conditioned intermediate features for variance regression. Conversely, the aleatoric branch's calibrated modeling of data-dependent variability provides an additional normalization signal to the backbone, helping the epistemic branch distinguish meaningful distributional deviations from sample-specific noise. Overall, these results confirm that CUPID's two-branch design forms a synergistic training mechanism, with the joint model consistently producing more reliable and discriminative uncertainty estimates than either branch trained in isolation.

Table 5: Misclassification detection performance on GLV2 (Joint vs. separate branches).

| Model | Aleatoric | | | Epistemic | | |
| --- | --- | --- | --- | --- | --- | --- |
| | AUC ($\uparrow$) | AURC ($\downarrow$) | Spearman ($\uparrow$) | AUC ($\uparrow$) | AURC ($\downarrow$) | Spearman ($\uparrow$) |
| Joint | **0.870 ± 0.002** | **0.018 ± 0.001** | **0.941 ± 0.004** | **0.769 ± 0.015** | **0.034 ± 0.002** | **0.701 ± 0.051** |
| Seperate | 0.863 ± 0.003 | 0.019 ± 0.001 | 0.899 ± 0.035 | 0.744 ± 0.017 | 0.043 ± 0.005 | 0.699 ± 0.014 |

Table 6: OOD detection performance (Joint vs. separate branches).

| Model | | PAPILA | | ACRIMA | | CIFAR10 | |
| --- | --- | --- | --- | --- | --- | --- | --- |
| | | AUC($\uparrow$) | AUPR($\uparrow$) | AUC($\uparrow$) | AUPR($\uparrow$) | AUC($\uparrow$) | AUPR($\uparrow$) |
| Alea. | Joint | 0.379 ± 0.027 | 0.333 ± 0.007 | 0.717 ± 0.029 | 0.661 ± 0.027 | **0.983 ± 0.005** | **0.998 ± 0.001** |
| Alea. | Seperate | 0.508 ± 0.097 | 0.385 ± 0.052 | 0.739 ± 0.071 | 0.661 ± 0.066 | 0.969 ± 0.027 | 0.995 ± 0.005 |
| Epis. | Joint | **0.877 ± 0.032** | **0.854 ± 0.027** | **0.978 ± 0.010** | **0.984 ± 0.007** | 0.898 ± 0.054 | 0.991 ± 0.005 |
| Epis. | Seperate | 0.771 ± 0.051 | 0.707 ± 0.073 | 0.972 ± 0.010 | 0.978 ± 0.009 | 0.844 ± 0.049 | 0.986 ± 0.005 |

## 5 CONCLUSION

CUPID acts as a versatile and lightweight plug-in module for joint aleatoric and epistemic uncertainty estimation. Without modifying or retraining the base model, CUPID can be inserted at any intermediate layer to reveal hidden sources of uncertainty across a wide range of tasks and datasets. Through comprehensive experiments on image classification and super-resolution, we show that CUPID consistently produces reliable uncertainty estimates. Beyond performance, our analysis of uncertainty propagation offers new insights into the internal behavior of neural networks. CUPID thus contributes both a practical tool for trustworthy AI and a conceptual lens for understanding model confidence.

ACKNOWLEDGMENTS

This research is supported by the Ministry of Education, Singapore (Grant IDs: RG15/23 and LKCMedicine Start up Grant) and the Centre of AI in Medicine (C-AIM), Nanyang Technological University.

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

## A  THE USE OF LARGE LANGUAGE MODELS (LLMs)

We used large language models to aid in polishing the manuscript. Specifically, an LLM was employed to refine grammar when necessary. All conceptual contributions, experiment design, data analysis, and interpretation were performed by the authors, with LLM support limited to language refinement as described in the paper.

## B  THEORETICAL ANALYSIS OF CUPID EPISTEMIC UNCERTAINTY

In this section, we provide a theoretical analysis of the CUPID Epistemic Uncertainty, focusing on its sensitivity to perturbations and the magnitude of deviation. We begin by reviewing the relevant notation and definitions.

Let $\mathbf{m}_l = B_l(\mathbf{x}) \in \mathbb{R}^d$ be the feature representation at the $l$-th layer of a sample $\mathbf{x}$, and let $F_l : \mathbb{R}^d \to \mathbb{Y}$ denote the downstream sub-network from this layer. The CUPID Reconstruction Branch produces a perturbed feature $\mathbf{m}'_l = C(\mathbf{m}_l)$, constrained to leave the final output nearly unchanged: $\|F_l(\mathbf{m}'_l) - F_l(\mathbf{m}_l)\| \le \epsilon$. The epistemic uncertainty is defined as the output deviation induced by this transformation:

$$U_{\text{epis}}(\mathbf{x}) := \|F_l(\mathbf{m}'_l) - F_l(\mathbf{m}_l)\|. \tag{17}$$

**Theorem 1** (Sensitivity & Deviation Driven Approximation of Epistemic Uncertainty). *Assume that $F_l$ is locally differentiable at $\mathbf{m}_l$, and let $\Delta\mathbf{m}_l = \mathbf{m}'_l - \mathbf{m}_l$ be the reconstruction perturbation. Then, under a first-order Taylor approximation:*

$$U_{epis}(\mathbf{x}) \approx \|J_{F_l}(\mathbf{m}_l) \cdot \Delta\mathbf{m}_l\|, \tag{18}$$

*where $J_{F_l}(\mathbf{m}_l)$ is the Jacobian of $F_l$ evaluated at $\mathbf{m}_l$.*

This result implies that the epistemic uncertainty estimated by CUPID is determined by two critical factors: the local sensitivity of the network (captured by the Jacobian) and the feature-space deviation introduced by the reconstruction:

$$U_{\text{epis}}(\mathbf{x}) \propto \text{Sensitivity} \times \text{Deviation}. \tag{19}$$

In the following sections, we further analyze the reliability of CUPID's epistemic uncertainty estimation, with a detailed discussion on the roles of sensitivity and deviation.

### B.1  DEVIATION-BASED ESTIMATION OF EPISTEMIC UNCERTAINTY

Epistemic uncertainty arises from a model's incomplete knowledge of its parameters, typically due to limited or insufficient training data. A common approach to quantifying this type of uncertainty is to approximate the posterior distribution over model parameters and evaluate the variability in predictions induced by sampling from this distribution. Proposition 3.1 in Wang & Ji (2024) formalizes this idea by showing that, under regularity conditions and in the large-data limit, the posterior distribution $p(\boldsymbol{\theta} \mid \mathcal{D})$ converges to a multivariate Gaussian centered at the maximum a posteriori estimate $\boldsymbol{\theta}^*$:

$$p(\boldsymbol{\theta} \mid \mathcal{D}) \approx \mathcal{N}(\boldsymbol{\theta}^*, \sigma^2\mathbf{I}). \tag{20}$$

This Gaussian approximation justifies modeling epistemic uncertainty through perturbations around model's parameter $\boldsymbol{\theta}^*$. Building upon this idea, we propose an alternative formulation that characterizes epistemic uncertainty through structured deviations in the input space. Rather than introducing randomness in the parameter space, we identify directions in the input domain along which the model output remains stable under the current parameterization. This provides a deterministic and geometrically interpretable estimate of epistemic uncertainty, formalized in the following proposition:

**Proposition 1.** *Let $f(\mathbf{x}, \boldsymbol{\theta}^*)$ be a neural network with fixed parameters $\boldsymbol{\theta}^*$. Define the deviation $\Delta\mathbf{x}^*$ as the solution to the following optimization problem:*

$$\begin{aligned} \Delta\mathbf{x}^* = \arg\max_{\Delta\mathbf{x}} \quad & \|\Delta\mathbf{x}\| \\ \text{subject to} \quad & \|f(\mathbf{x} + \Delta\mathbf{x}, \boldsymbol{\theta}^*) - f(\mathbf{x}, \boldsymbol{\theta}^*)\| \le \delta, \end{aligned} \tag{21}$$

*for a small tolerance $\delta > 0$. Then, there exists a parameter perturbation $\Delta\boldsymbol{\theta}$ such that:*

$$f(\mathbf{x} + \Delta\mathbf{x}^*, \boldsymbol{\theta}^*) = f(\mathbf{x}, \boldsymbol{\theta}^* + \Delta\boldsymbol{\theta}). \tag{22}$$

This result shows that the deviation $\Delta\mathbf{x}^*$, which is constrained to preserve the output, can approximate the effect of a parameter perturbation. Hence, the deviation acts as a proxy for epistemic uncertainty, enabling its deterministic and input-dependent estimation.

**Proof of Proposition 1.** To justify the equivalence, we consider first-order Taylor approximations of the model with respect to both the input and the parameters:

$$\begin{aligned} f(\mathbf{x} + \Delta\mathbf{x}^*, \boldsymbol{\theta}^*) &\approx f(\mathbf{x}, \boldsymbol{\theta}^*) + J_{\mathbf{x}} \cdot \Delta\mathbf{x}^*, \\ f(\mathbf{x}, \boldsymbol{\theta}^* + \Delta\boldsymbol{\theta}) &\approx f(\mathbf{x}, \boldsymbol{\theta}^*) + J_{\boldsymbol{\theta}} \cdot \Delta\boldsymbol{\theta}, \end{aligned} \tag{23}$$

where $J_{\mathbf{x}} = \frac{\partial f(\mathbf{x}, \boldsymbol{\theta}^*)}{\partial \mathbf{x}}$ and $J_{\boldsymbol{\theta}} = \frac{\partial f(\mathbf{x}, \boldsymbol{\theta}^*)}{\partial \boldsymbol{\theta}}$.

Since $\Delta\mathbf{x}^*$ is chosen such that $f(\mathbf{x} + \Delta\mathbf{x}^*, \boldsymbol{\theta}^*) \approx f(\mathbf{x}, \boldsymbol{\theta}^*)$, we have:

$$J_{\mathbf{x}} \cdot \Delta\mathbf{x}^* \approx 0. \tag{24}$$

We now seek a $\Delta\boldsymbol{\theta}$ such that:

$$\begin{aligned} f(\mathbf{x}, \boldsymbol{\theta}^* + \Delta\boldsymbol{\theta}) &= f(\mathbf{x} + \Delta\mathbf{x}^*, \boldsymbol{\theta}^*) \approx f(\mathbf{x}, \boldsymbol{\theta}^*) \\ &\Rightarrow \quad J_{\boldsymbol{\theta}} \cdot \Delta\boldsymbol{\theta} \approx 0. \end{aligned} \tag{25}$$

Thus, any $\Delta\boldsymbol{\theta}$ in the null space of $J_{\boldsymbol{\theta}}$ satisfies this condition. In particular, we can construct such a $\Delta\boldsymbol{\theta}$ by perturbing the first-layer weights.

Let $\boldsymbol{\theta}_1$ denote the weights of the first layer, and consider the linear transformation:

$$f^{(1)}(\mathbf{x}, \boldsymbol{\theta}_1) = \sigma(\boldsymbol{\theta}_1^\top \mathbf{x}), \tag{26}$$

where $\sigma$ is the activation function. To preserve the first-layer output under the input deviation $\Delta\mathbf{x}^*$, we require:

$$(\boldsymbol{\theta}_1 + \Delta\boldsymbol{\theta}_1)^\top \mathbf{x} = \boldsymbol{\theta}_1^\top(\mathbf{x} + \Delta\mathbf{x}^*) \quad \Rightarrow \quad \Delta\boldsymbol{\theta}_1^\top \mathbf{x} = \boldsymbol{\theta}_1^\top \Delta\mathbf{x}^*. \tag{27}$$

Assuming $x_k \neq 0$, this condition is satisfied by:

$$\Delta\theta_{1,kj} = \theta_{1,kj} \cdot \frac{\Delta x_k^*}{x_k}. \tag{28}$$

This construction ensures that:

$$f^{(1)}(\mathbf{x} + \Delta\mathbf{x}^*, \boldsymbol{\theta}_1) = f^{(1)}(\mathbf{x}, \boldsymbol{\theta}_1 + \Delta\boldsymbol{\theta}_1), \tag{29}$$

and under mild smoothness conditions on $\sigma$ and subsequent layers, this local equivalence propagates to the full network output. $\square$

**Extension to Intermediate Features.** Although the preceding formulation is derived based on input-level perturbations, the underlying reasoning naturally extends to internal feature representations within the network. Specifically, consider an intermediate feature vector $\mathbf{m}_l = B_l(\mathbf{x}) \in \mathbb{R}^d$ at layer $l$, where $B_l$ denotes the sub-network up to layer $l$, and let $F_l : \mathbb{R}^d \to \mathbb{Y}$ be the sub-network from layer $l$ to the output.

We define the deviation $\Delta\mathbf{m}_l^*$ in the feature space as the solution to the following optimization problem:

$$\begin{aligned} \Delta\mathbf{m}_l^* = \arg\max_{\Delta\mathbf{m}_l} \quad &\|\Delta\mathbf{m}_l\| \\ \text{subject to} \quad &\|F_l(\mathbf{m}_l + \Delta\mathbf{m}_l) - F_l(\mathbf{m}_l)\| \leq \delta, \end{aligned} \tag{30}$$

for a small tolerance $\delta > 0$. This formulation mirrors the input-level case and enables direct manipulation of internal activations while preserving output consistency.

By optimizing deviations in intermediate layers, our method generalizes naturally to feature-based or modular architectures. It is particularly useful in scenarios where inputs are fixed or uninterpretable, but internal representations can be explicitly perturbed and interpreted. This makes our epistemic uncertainty estimation applicable across a broader range of neural architectures and analysis settings.

## B.2 SENSITIVITY AS AN INDICATOR OF EPISTEMIC UNCERTAINTY

While the previous subsection establishes a connection between input perturbations and equivalent parameter shifts, this section explores how sensitivity (quantified via gradients) can serve as a direct and meaningful indicator of epistemic uncertainty. Specifically, we argue that the magnitude of the gradient of the model output with respect to its parameters reflects the model's familiarity with a given input.

Proposition 3.4 in Wang & Ji (2024) shows that, for inputs close to the training distribution, a sufficiently trained model exhibits vanishing gradients with respect to its parameters. More precisely, in a neighborhood $\mathcal{N}(\mathbf{x}_0)$ around an in-distribution point $\mathbf{x}_0$, the gradient satisfies:

$$\nabla_{\boldsymbol{\theta}} f(\mathbf{x}, \boldsymbol{\theta}^*) = 0, \quad \forall \mathbf{x} \in \mathcal{N}(\mathbf{x}_0). \tag{31}$$

This result motivates the use of the gradient norm as a proxy for epistemic uncertainty: for inputs the model is confident about, small or vanishing gradients imply stability under parameter perturbations; conversely, large gradients indicate potential epistemic mismatch.

Building on this, we propose an alternative formulation in which the perturbation $\Delta\mathbf{x}^*$ is not sampled randomly but is instead optimized under a constraint that the output remains stable. Rather than exploring the entire local input neighborhood indiscriminately, we restrict the perturbation to directions that preserve the output, leading to a boundary-aware sensitivity measure. This motivates the following proposition:

**Proposition 2.** *Let $f(\mathbf{x}, \boldsymbol{\theta})$ be a neural network with fixed parameters $\boldsymbol{\theta}^*$, and define the perturbation $\Delta\mathbf{x} = \|C(\mathbf{x}) - \mathbf{x}\|$ by:*

$$\Delta\mathbf{x}^* = \arg\max_{\Delta\mathbf{x}} \quad \|\Delta\mathbf{x}\|$$
$$\text{subject to} \quad \|f(\mathbf{x} + \Delta\mathbf{x}, \boldsymbol{\theta}^*) - f(\mathbf{x}, \boldsymbol{\theta}^*)\| \leq \delta, \tag{32}$$

*for a small tolerance $\delta > 0$. Then, if $\mathbf{x}$ is in-distribution and the model is well-trained, we have:*

$$\nabla_{\boldsymbol{\theta}} f(\mathbf{x}, \boldsymbol{\theta}^*) = 0, \quad \text{and} \quad \|\nabla_{\boldsymbol{\theta}} f(\mathbf{x} + \Delta\mathbf{x}^*, \boldsymbol{\theta}^*)\| \to 0. \tag{33}$$

This result can be interpreted as an extension of Proposition 3.4 from Wang & Ji (2024). While the original proposition attributes vanishing parameter gradients to the convergence of the posterior around in-distribution inputs, our formulation adds an output-preservation constraint. This constraint restricts the perturbation to lie within a local iso-response surface—i.e., the subspace where the output remains stable under the fixed model parameters. If a large perturbation $\Delta\mathbf{x}^*$ still maintains the output within $\delta$, the model is considered epistemically confident around $\mathbf{x}$. For in-distribution data $\mathbf{x}$, a well-trained model satisfies the first-order optimality condition:

$$\nabla_{\boldsymbol{\theta}} f(\mathbf{x}, \boldsymbol{\theta}^*) = 0. \tag{34}$$

Now, consider the perturbation $\Delta\mathbf{x}^*$ that maximizes $\|\Delta\mathbf{x}\|$ while ensuring $f(\mathbf{x} + \Delta\mathbf{x}, \boldsymbol{\theta}^*) \approx f(\mathbf{x}, \boldsymbol{\theta}^*)$. Since the output does not significantly change, we infer that the model's prediction remains on the same confidence surface. Assuming smoothness of $f$, we can expand $f(\mathbf{x} + \Delta\mathbf{x}, \boldsymbol{\theta})$ in a Taylor series around $\mathbf{x}$, and observe that for sufficiently small $\delta$, the leading-order change in the parameter gradient at $\mathbf{x} + \Delta\mathbf{x}^*$ is also small:

$$\nabla_{\boldsymbol{\theta}} f(\mathbf{x} + \Delta\mathbf{x}^*, \boldsymbol{\theta}^*) \to 0 \quad \text{as} \quad \delta \to 0. \tag{35}$$

This implies that the model remains insensitive to parameter perturbations in the vicinity of $\mathbf{x} + \Delta\mathbf{x}^*$, confirming the epistemic confidence around that region.

## C TABLAR EXAMPLE

**Dataset** We evaluate our method on the Covertype dataset, a classical structured-data benchmark provided by the UCI Machine Learning Repository (Blackard, 1998). The dataset contains 581012 samples with 54 continuous and binary features, representing cartographic variables such as elevation, slope, and soil type. Each sample corresponds to a 30m × 30m cell of forest cover in the Roosevelt National Forest of northern Colorado. The classification task is to predict the forest cover type, which falls into one of seven possible categories (multi-class setting). We randomly split the dataset into 80% training and 20% testing sets. Since this dataset is tabular rather than image-based, it provides a complementary evaluation to the image-centric experiments in the main paper, highlighting the generality of our proposed uncertainty estimation framework.

Table 7: Hyperparameters for tabular examples.

| Hyperparameters | Predicted Model | CUPID |
|---|---|---|
| Epoch | 50 | 50 |
| Batch size | 256 | 256 |
| Learning rate | 0.001 | 0.0001 |
| $\lambda_1$ | / | 0.001 |
| $\lambda_2$ | / | 0.01 |

Table 8: Performance of tabular examples.

| Method | CUPID Alea. | CUPID Epis. | MC Dropout |
|---|---|---|---|
| AUC($\uparrow$) | $0.965 \pm 0.000$ | - | - |
| Accuracy($\uparrow$) | $0.837 \pm 0.000$ | - | - |
| AUC ($\uparrow$) | $\mathbf{0.769 \pm 0.006}$ | $0.688 \pm 0.005$ | $0.563 \pm 0.001$ |
| AURC ($\downarrow$) | $\mathbf{0.060 \pm 0.001}$ | $0.088 \pm 0.002$ | $0.138 \pm 0.000$ |
| Spearman ($\uparrow$) | $\mathbf{0.812 \pm 0.017}$ | $0.627 \pm 0.012$ | $0.365 \pm 0.001$ |

**Model** For the classification task, we adopt a Multi-Layer Perceptron (MLP) consisting of four fully connected layers. Each layer is followed by a Sigmoid activation function, and dropout layers are applied between hidden layers to prevent overfitting. The CUPID model follows the MLP architecture as the base classifier and is placed before the final Linear layer. The hyperparameters of the classification model and CUPID model on toy examples are shown in Table 7.

**Results** We conduct misclassification detection experiments on the dataset and compare the performance of CUPID with MC Dropout, using the same MLP baseline model. The results are presented in Table 8. As shown, CUPID achieves significantly better uncertainty estimation performance across all metrics. In particular, CUPID's aleatoric uncertainty yields an AUC of 0.769 and Spearman correlation of 0.812, indicating a strong correlation between uncertainty and prediction errors. In contrast, MC Dropout achieves lower AUC (0.563) and Spearman (0.365), reflecting less effective uncertainty estimates. Moreover, CUPID's epistemic uncertainty also outperforms MC Dropout, achieving a Spearman correlation of 0.627 versus 0.365 and demonstrating its ability to capture model uncertainty. These results validate the effectiveness of CUPID in estimating both aleatoric and epistemic uncertainties for misclassification detection in tabular data.

# D    SUPPLEMENTS FOR THE EXPERIMENTAL SETTING

All experiments are conducted on a workstation equipped with an NVIDIA GeForce RTX 4090 GPU. The software environment includes Python 3.9 and PyTorch 2.0.1.

## D.1    DETAILS OF THE TOY EXAMPLES

**Datasets** We constructed two synthetic datasets to evaluate our method under controlled settings. The dataset depicted in Fig. 1 (Left) of the main paper was generated using the following formulation for the target variable $y$:

$$
\begin{cases}
3\sin(0.8x) + 5.3 + \epsilon, \\
\quad \text{where } \epsilon \sim \mathcal{N}(0, \ 0.7), \quad x \in [5, \ 8) \cup [12, \ 14) \\
\\
3\sin(0.8x) + 5.7 + \epsilon, \\
\quad \text{where } \epsilon \sim \mathcal{N}(0, \ 0.3). \quad x \in [8, \ 12)
\end{cases}
\tag{36}
$$

Table 9: Hyperparameters for toy examples.

| Hyperparameters | Predicted Model | CUPID |
|---|---|---|
| Max epoch | 50 | 50 |
| Batch size | 16 | 8 |
| Learning rate | 0.001 | 0.001 |
| $\lambda_1$ | / | 0.001 |
| $\lambda_2$ | / | 0.01 |

And the dataset demonstrated in Fig. 1 (Right) is formulated as:

$$
\begin{cases}
3\sin(0.8x) + \sin(2x) + 1.3 + \epsilon, \\
\text{where } \epsilon \sim \mathcal{N}(0, 0.7), \quad x \in [5, 9) \\
\\
3\sin(0.8x) + \sin(2x) + 1.8 + \epsilon, \\
\text{where } \epsilon \sim \mathcal{N}(0, 0.2). \quad x \in [11, 13)
\end{cases}
\tag{37}
$$

**Model** The regression model used in the toy experiments is a three-layer MLP with sigmoid activation functions. To align with the structure of the predictive model, the CUPID model employed in this setting is also implemented as an MLP, following a similar yet simplified architecture compared to the version used in our main experiments. Specifically, the feature extractor component in this variant consists of only two blocks, reducing complexity while preserving core functionality. The CUPID model is integrated into the regression network by inserting it immediately before the final linear layer.

The hyperparameters of the regression and CUPID model on toy examples are shown in Table 9.

### D.2 DETAILS OF MEDICAL IMAGE CLASSIFICATION EXPERIMENTS

**Datasets** We evaluate our method on two widely used medical imaging datasets designed for classification tasks involving different modalities and diseases: Glaucoma-Light V2 (GLV2) and HAM10000. Visual examples are shown in Figure 5.

*GLV2* (Gulshan et al., 2016; Kiefer et al., 2022) is a large-scale fundus image dataset comprising 4,770 referable glaucoma (RG) and 4,770 non-referable glaucoma (NRG) images. The data is divided into training, validation, and test sets, each containing 4,000, 385, and 385 samples per class, respectively. This structure ensures a well-controlled evaluation setting with equal representation of both classes.

*HAM10000* (Tschandl et al., 2018) is a dermoscopic image dataset developed for the classification of pigmented skin lesions. It includes a diverse range of skin conditions and imaging settings. For this study, we follow the official split: 10,015 images for training, 193 for validation, and 1,512 for testing. Each image is centered on a lesion and captured under standardized lighting to minimize acquisition bias.

We quantify acquisition-related randomness using dataset-level pixel variance ("Noise") and signal-to-noise ratio (SNR), computed from the first- and second-order moments of all images (Sahoo et al., 2025). Higher noise and lower SNR indicate stronger imaging variability, which is a typical driver of aleatoric uncertainty. As shown in Table 10, GLV2 exhibits higher Noise (0.024 vs. 0.022) and markedly lower SNR (5.44 dB vs. 12.78 dB) than HAM10000, confirming that GLV2 contains more acquisition-induced variability and is therefore more AU-dominant. EU arises from limited or uneven data coverage. HAM10000 contains 7 diagnostic classes with substantial imbalance (142–6705 samples), forming a heterogeneous and sparsely supported feature space. In contrast, GLV2 consists of only 2 well-balanced classes. This structural difference implies that models trained on HAM10000 must learn more complex and uneven decision boundaries, resulting in higher epistemic uncertainty, while GLV2's simpler and balanced distribution yields lower EU.

**Predictive Base Model** $M$ The predictive model $M$ is implemented as a ResNet18 (He et al., 2016) architecture, a widely-used convolutional neural network known for its residual learning ca-

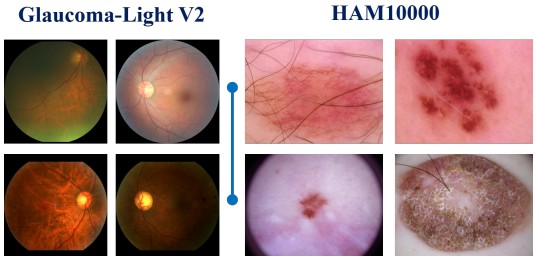

Figure 5: Data samples from GLV2 and HAM10000.

Table 10: Dataset characteristics indicating different sources of uncertainty.

| Dataset | Noise ($\uparrow$) | SNR ($\downarrow$) | Classes | Class balanced |
|---------|---------|---------|---------|----------------|
| GLV2 | 0.024 | 5.44 dB | 2 | True |
| HAM10000 | 0.022 | 12.78 dB | 7 | False |

pabilities. The network consists of 18 layers, including a $7 \times 7$ convolutional layer and max-pooling at the input, followed by four stages of residual blocks with increasing filter dimensions (64, 128, 256, and 512). Each residual block contains two $3 \times 3$ convolutions with batch normalization and ReLU activation, facilitating stable gradient flow in deeper networks. The model concludes with a global average pooling layer and a fully connected layer to output classification logits.

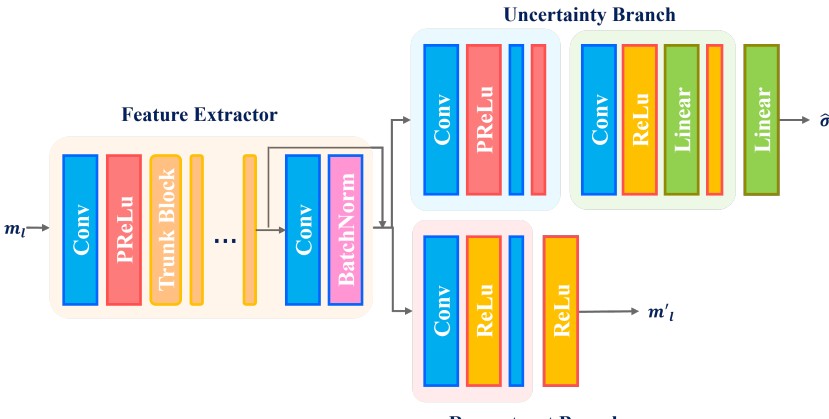

Figure 6: The structure of CUPID used in ResNet18 for the classification problem.

**CUPID Model** $C$    The architecture of CUPID used in these experiments is illustrated in Figure 6. The CUPID model consists of three functional components designed to estimate uncertainty and perturb latent representations in a structured way:

*Feature Extractor*: This module processes the intermediate feature map $\mathbf{m_l}$ using a series of trunk blocks, which adopt the Residual-in-Residual Dense Block (RRDB) structure (Wang et al., 2018b). A batch normalization layer follows the trunk blocks, yielding a refined latent representation that serves as a shared input to the subsequent branches.

*Uncertainty Branch*: This branch estimates aleatoric uncertainty by learning an uncertainty score. It begins with convolutional and PReLU layers to maintain spatial structure and introduce nonlinearity, followed by fully connected layers and ReLU activations to project the features into uncertainty value. This score captures the noise-related variability in the data that affects model predictions. Linear layer is chosen as the last layer because we model the log-variance $\mathbf{s_n} = \log(\hat{\boldsymbol{\sigma}}_n^2)$ instead of variance directly.

Table 11: Hyperparameters for medical image classification experiments.

| Hyperparameters | GLV2 | | HAM10000 | |
|---|---|---|---|---|
| | **Predicted** | **CUPID** | **Predicted** | **CUPID** |
| Max epoch | 10 | 15 | 10 | 15 |
| Batch size | 4 | 4 | 8 | 4 |
| Learning rate | 0.00001 | 0.00001 | 0.0001 | 0.00001 |
| Decay step size | 1 | 1 | 4.5 | 1 |
| Decay $\gamma$ | 0.8 | 0.9 | 0.75 | 0.9 |
| $\lambda_1$ | / | 0.01 | / | 0.01 |
| $\lambda_2$ | / | 0.009 | / | 0.009 |

Table 12: Mean accuracy and AUC for each classification model on GLV2 and HAM10000 datasets.

| Method | GLV2 | | HAM10000 | |
|---|---|---|---|---|
| | **AUC($\uparrow$)** | **Acc($\uparrow$)** | **AUC($\uparrow$)** | **Acc($\uparrow$)** |
| CUPID/IGRUE/Rate-in | $0.970 \pm 0.000$ | $0.909 \pm 0.000$ | $0.952 \pm 0.000$ | $0.821 \pm 0.000$ |
| MC Dropout | $0.979 \pm 0.000$ | $0.921 \pm 0.000$ | $0.946 \pm 0.000$ | $0.765 \pm 0.002$ |
| PostNet | $0.789 \pm 0.004$ | $0.718 \pm 0.010$ | $0.865 \pm 0.002$ | $0.664 \pm 0.008$ |
| BNN | $0.966 \pm 0.005$ | $0.914 \pm 0.004$ | $0.929 \pm 0.004$ | $0.742 \pm 0.011$ |
| DEC | $0.875 \pm 0.002$ | $0.878 \pm 0.003$ | $0.922 \pm 0.014$ | $0.768 \pm 0.009$ |

*Reconstruction Branch*: This module reconstructs a perturbed version $\mathbf{m}_1'$ of the original intermediate feature map. It consists of two convolutional layers and ReLU activations, aiming to modify the feature representation while preserving its overall structure. The reconstructed feature is then fed back into the remaining layers of the predictive model $M$ to measure epistemic uncertainty through output deviation.

Empirically, we observe that the number of layers in each component has a limited impact on overall performance. CUPID is inserted between residual blocks of ResNet18. Since ResNet employs ReLU activations after each residual block, it is critical that the final layer of CUPID's Reconstruction Branch also uses a compatible activation. Failing to match activation functions can hinder training due to mismatched feature distributions.

**Training and Classification Results** The hyperparameters for both the predictive model and the CUPID module were optimized through random search and are detailed in Table 11.

We benchmark our approach against several widely adopted uncertainty estimation techniques: Rate-in (Zeevi et al., 2025), Monte Carlo Dropout (MC Dropout) (Gal & Ghahramani, 2016), Posterior Network (PostNet) (Charpentier et al., 2020), Deep Evidential Classification (DEC) (Sensoy et al., 2018), Bayesian Neural Network (BNN), and IGRUE (Wang et al., 2023). All models are trained under identical data splits for GLV2 and HAM10000 to enable robust comparisons. All experiments were conducted using ResNet18 as the backbone to ensure a fair and consistent comparison across all methods.

Table 12 reports the AUC and accuracy results. All models were trained with early stopping based on the validation loss to ensure optimal generalization performance. CUPID, Rate-in and IGRUE are applied on the same base model. They both operate on the intermediate features of the pretrained model without modifying the parameters of the original classifier. This design ensures that high classification performance can be maintained while gaining reliable uncertainty estimates.

MC Dropout was implemented by adding a dropout layer with a drop rate of 0.03 at the end of the ResNet18 model. During inference, uncertainty was estimated using ten stochastic forward passes. PostNet, BNN, and DEC required training from scratch and were implemented using the ResNet18 backbone for consistency. These models were trained for up to 200 epochs with early stopping (patience of 10 epochs), and hyperparameters were tuned to achieve optimal classification performance.

**Glaucoma-Light V2**     **PAPILA**     **ACRIMA**     **CIFAR-10**

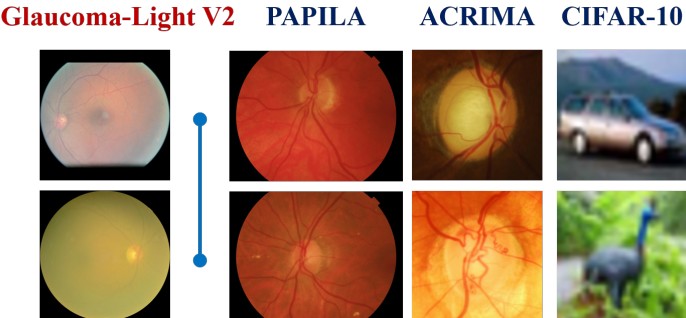

Figure 7: Data samples of ID and OOD datasets.

## D.3 DETAILS OF THE OOD DETECTION EXPERIMENTS

**Datasets**   We employ four datasets to evaluate the performance of out-of-distribution (OOD) detection. GLV2 serves as the in-distribution (ID) dataset, selected for its large scale, high quality, and balanced class distribution. Its diversity and size make it well-suited for learning the target distribution in a supervised setting. The remaining three datasets (ACRIMA (Kovalyk et al., 2022), PAPILA (Kovalyk et al., 2022), and CIFAR-10 (Krizhevsky & Hinton, 2009)) are treated as OOD datasets. These datasets differ from GLV2 in varying degrees, allowing us to assess how well each model estimates uncertainty under different levels of domain shift. The data samples of ID and OOD datasets are shown in Figure 7. A detailed description of each dataset is provided below:

*PAPILA*. This dataset contains high-resolution (2,576 × 1,934) fundus images from both eyes of 244 subjects, totaling 488 images. It includes three categories: healthy, glaucoma, and suspicious. Compared to GLV2, PAPILA images tend to be darker, redder, and lower in contrast. Furthermore, due to the fixed input size requirement (512 × 512), the images undergo deformation during resizing, adding an additional distributional shift.

*ACRIMA*. ACRIMA includes 705 labeled fundus images (396 glaucomatous and 309 normal). Unlike GLV2, the images in ACRIMA are preprocessed by cropping around the optic disc to emphasize clinically relevant features. This alteration introduces a distributional shift focused on spatial and contextual features.

*CIFAR-10*. CIFAR-10 is a natural image dataset designed for object classification across 10 distinct categories: airplane, automobile, bird, cat, deer, dog, frog, horse, ship, and truck. It contains 50,000 training images and 10,000 test images with a resolution of 32 × 32. Its unrelated domain makes it a strong test for extreme OOD scenarios.

**Model and Testing Protocol**   The models evaluated in this experiment are the same as those used in the misclassification detection task, including our proposed CUPID, Rate-in, MC Dropout, Post-Net, BNN, DEC, and IGRUE. All models were trained exclusively on the GLV2 dataset and then tested on the ID (GLV2) and three OOD datasets (PAPILA, ACRIMA, and CIFAR-10). To ensure consistency, all input images were resized to 512 × 512 pixels. For CIFAR-10, we retained their original training, validation, and test splits. Since PAPILA and ACRIMA lack predefined splits and are relatively small in size, the entire datasets were used for testing.

## D.4 DETAILS OF THE SINGLE IMAGE SUPER RESOLUTION EXPERIMENTS

**Datasets for Super-Resolution**   We trained the ESRGAN and CUPID on DIV2K and evaluated our method on four widely-used benchmark datasets for single image super-resolution under a ×4 scale setting with bicubic downsampling.

*DIV2K* (Agustsson & Timofte, 2017) serves as the primary training dataset and consists of 800 high-resolution (2K) images across diverse scenes, offering rich textures and fine details suitable for learning robust SR models.

Table 13: Hyperparameters for super-resolution.

| Hyperparameters | Predicted Model | CUPID |
|---|---|---|
| Max iteration | 300000 | 300000 |
| Batch size | 16 | 16 |
| Learning rate | 0.0001 | 0.00001 |
| $\lambda_1$ | / | 0.0001 |
| $\lambda_2$ | / | 0.01 |

*Set5* (Bevilacqua et al., 2012) includes 5 classical natural images commonly used for SR benchmarking. The dataset covers various object categories such as animals, architecture, and people, allowing for controlled evaluation in small-scale settings.

*Set14* (Zeyde et al., 2010) comprises 14 images with greater diversity in scene content and degradation types compared to Set5, including urban structures, facial portraits, and natural landscapes.

*BSDS100* (Martin et al., 2001), a subset of the Berkeley Segmentation Dataset, contains 100 test images with a wide variety of textures and fine structures. It is widely adopted to assess generalization and edge reconstruction quality in SR tasks.

*IXI* (Biomedical Image Analysis Group, Imperial College London, 2022) is a publicly available brain MRI dataset collected from three hospitals in London: Guy's Hospital, Hammersmith Hospital, and the Institute of Psychiatry. It contains over 600 subjects and includes various imaging modalities such as T1-weighted, T2-weighted, proton density (PD), and diffusion-weighted images. The dataset covers a broad age range and provides valuable anatomical diversity, making it a widely used resource for developing and evaluating medical image processing and machine learning algorithms, particularly in brain structure analysis and image reconstruction tasks.

**ESRGAN Backbone** The Enhanced Super-Resolution Generative Adversarial Network (ESRGAN) is a widely adopted architecture for perceptual single-image super-resolution, known for its ability to produce high-fidelity reconstructions with visually realistic textures. It builds upon the original SRGAN framework by incorporating several architectural improvements aimed at enhancing perceptual quality and training stability.

ESRGAN comprises three primary components: a generator, a discriminator, and a perceptual loss module. The generator is constructed using a deep convolutional architecture based on Residual-in-Residual Dense Blocks (RRDBs), which synergize the advantages of residual learning and dense connectivity. Unlike traditional residual blocks, RRDBs omit batch normalization layers to avoid artifacts and enable more stable training. These blocks allow for efficient feature propagation and better gradient flow, enabling the network to preserve fine-grained textures across deep layers. Multiple RRDBs are stacked to form the feature extraction backbone, followed by a series of upsampling blocks that progressively increase the spatial resolution of the image. These upsampling layers are placed toward the end of the network to reduce computational cost during earlier stages. A final convolutional layer refines the output to produce the high-resolution image.

In our experiments, we evaluate the placement of the CUPID module by inserting it either before or after the upsampling blocks. This setup allows us to study how the model's uncertainty estimation capabilities vary depending on its position relative to the reconstruction pipeline.

**CUPID Model $C$ for Super-Resolution** The CUPID architecture for image super-resolution largely follows the classification variant but adapts to the demands of pixel-level precision. Batch normalization is omitted in the feature extractor to preserve local image statistics, while the Uncertainty Branch incorporates convolutional layers with Leaky ReLU and a multi-stage upsampling module, producing a spatial uncertainty map aligned with the super-resolved image for aleatoric uncertainty estimation. In parallel, the Reconstruction Branch employs convolution and Leaky ReLU to reconstruct intermediate features, enhancing compatibility with the ESRGAN backbone and supporting epistemic uncertainty estimation.

Table 14: PSNR and SSIM for ESRGAN model.

| Dataset | SSIM(↑) | PSNR(↑) |
|---|---|---|
| Set5 | 0.828 | 28.533 |
| Set14 | 0.684 | 24.865 |
| BSDS100 | 0.631 | 23.539 |
| IXI | 0.806 | 33.142 |

**Training and Super-Resolution Results** The hyperparameters for both the predictive model and the proposed CUPID module were optimized using random search and are summarized in Table 13. All models were trained using an early stopping strategy based on validation loss to prevent overfitting and ensure optimal generalization. We compare CUPID against five representative uncertainty estimation baselines. BayesCap (Upadhyay et al., 2022) explicitly models the uncertainty by reconstructing an output distribution. It shares the same training protocol and predictive backbone (ESRGAN) as CUPID, allowing for a fair comparison.

*In-rotate* and *in-noise* estimate uncertainty by analyzing output variation under input transformations. *In-rotate* (Mi et al., 2022) applies four 90-degree rotations to the input image and computes the output variance as an uncertainty measure. *In-noise* injects 0.005% Gaussian noise into the input image and repeats the inference ten times to estimate uncertainty based on output variation (Wang et al., 2018a).

*Med-noise* and *med-dropout* (Mi et al., 2022) assess uncertainty through perturbations in the latent space. *Med-noise* adds 0.005% Gaussian noise to the intermediate feature map and repeats inference ten times. *Med-dropout* introduces an additional dropout layer with a drop probability of 0.3, also using ten repeated forward passes during testing.

The super-resolution performance of the baseline ESRGAN modelare reported in Table 14.

# E  OTHER EXPERIMENTS RESULTS

## E.1  HYPERPARAMETER EXPERIMENTS ON CUPID LOCATION

Table 15 shows the results of inserting CUPID at different intermediate layers. The results demonstrate a clear trend: aleatoric uncertainty is more accurately estimated when CUPID is placed closer to the output layer, while epistemic uncertainty benefits from earlier placements within the network.

Table 15: Results of inserting CUPID at different intermediate layers. Best-performing results for each metric are highlighted in bold.

| Class Position | GLV2 | | | HAM10000 | | |
|---|---|---|---|---|---|---|
| | AUC (↑) | AURC (↓) | Spearman (↑) | AUC (↑) | AURC (↓) | Spearman (↑) |
| Aleatoric (4) | **0.870 ± 0.002** | **0.018 ± 0.001** | **0.941 ± 0.004** | **0.769 ± 0.023** | **0.067 ± 0.007** | **0.722 ± 0.014** |
| Aleatoric (3) | 0.843 ± 0.008 | 0.022 ± 0.001 | 0.851 ± 0.006 | 0.751 ± 0.004 | 0.072 ± 0.003 | 0.624 ± 0.024 |
| Aleatoric (2) | 0.805 ± 0.014 | 0.026 ± 0.001 | 0.772 ± 0.019 | 0.749 ± 0.011 | 0.103 ± 0.006 | 0.675 ± 0.021 |
| Epistemic (4) | 0.769 ± 0.015 | 0.034 ± 0.002 | 0.701 ± 0.051 | 0.855 ± 0.006 | 0.047 ± 0.001 | **0.907 ± 0.001** |
| Epistemic (3) | 0.786 ± 0.003 | 0.033 ± 0.004 | 0.696 ± 0.015 | 0.869 ± 0.005 | **0.045 ± 0.000** | 0.898 ± 0.004 |
| Epistemic (2) | **0.789 ± 0.017** | **0.031 ± 0.001** | **0.717 ± 0.006** | **0.901 ± 0.004** | 0.058 ± 0.001 | 0.888 ± 0.005 |
| **SR Position** | **Set5** | | | **Set14** | | |
| | Pearson (↑) | AUSE (↓) | UCE (↓) | Pearson (↑) | AUSE (↓) | UCE (↓) |
| Aleatoric (A) | **0.560 ± 0.001** | **0.009 ± 0.000** | **0.034 ± 0.007** | **0.569 ± 0.001** | **0.011 ± 0.000** | **0.017 ± 0.008** |
| Aleatoric (B) | 0.528 ± 0.006 | 0.010 ± 0.000 | 0.045 ± 0.018 | 0.527 ± 0.002 | 0.012 ± 0.000 | 0.049 ± 0.005 |
| Epistemic (A) | 0.258 ± 0.005 | 0.026 ± 0.000 | 0.320 ± 0.004 | 0.327 ± 0.005 | 0.026 ± 0.000 | 0.231 ± 0.005 |
| Epistemic (B) | **0.416 ± 0.004** | **0.018 ± 0.001** | **0.266 ± 0.007** | **0.449 ± 0.005** | **0.019 ± 0.000** | **0.226 ± 0.003** |

Table 16: Uncertainty calibration results for misclassification (GLV2) and OOD detection (PAPILA, ACRIMA, CIFAR-10)

| Method | GLV2 | | PAPILA | ACRIMA | CIFAR10 |
| | rAULC (↑) | UCE (↓) | | OOD-UCE (↓) | |
|---|---|---|---|---|---|
| CUPID Alea. | **0.840 ± 0.003** | 0.042 ± 0.004 | 0.308 ± 0.026 | 0.226 ± 0.079 | 0.273 ± 0.004 |
| CUPID Epis. | 0.649 ± 0.031 | **0.033 ± 0.008** | **0.115 ± 0.016** | 0.240 ± 0.006 | 0.633 ± 0.017 |
| MC Dropout | 0.682 ± 0.009 | 0.052 ± 0.019 | 0.257 ± 0.001 | 0.307 ± 0.045 | 0.699 ± 0.021 |
| Rate-in | 0.762 ± 0.011 | 0.038 ± 0.009 | 0.309 ± 0.009 | 0.411 ± 0.008 | 0.826 ± 0.019 |
| IGRUE | 0.375 ± 0.024 | 0.145 ± 0.012 | 0.154 ± 0.046 | 0.252 ± 0.016 | 0.596 ± 0.010 |
| PostNet Alea. | 0.419 ± 0.019 | 0.166 ± 0.031 | 0.179 ± 0.025 | 0.285 ± 0.036 | 0.500 ± 0.125 |
| PostNet Epis. | 0.184 ± 0.066 | 0.254 ± 0.013 | 0.182 ± 0.104 | 0.289 ± 0.089 | 0.726 ± 0.082 |
| BNN | 0.747 ± 0.017 | 0.050 ± 0.007 | 0.268 ± 0.010 | 0.332 ± 0.012 | 0.705 ± 0.008 |
| DEC | -0.627 ± 0.056 | 0.417 ± 0.039 | 0.256 ± 0.027 | **0.198 ± 0.010** | **0.248 ± 0.008** |

### E.2 UNCERTAINTY CALIBRATION RESULTS FOR MISCLASSIFICATION AND OOD

Table 16 summarizes uncertainty calibration performance for both misclassification and OOD settings. CUPID consistently ranks first or second across all metrics, demonstrating strong uncertainty–error correlation and stable calibration. On GLV2, CUPID Aleatoric achieves the highest rAULC (0.840), while CUPID Epistemic obtains the best UCE (0.033), indicating excellent uncertainty ranking and calibration. For OOD detection, CUPID achieves the lowest OOD-UCE on PAPILA (0.115) and competitive results on ACRIMA (0.226 and 0.240). Notably, DEC attains the best OOD-UCE on ACRIMA (0.198), but this comes at the cost of severely degraded misclassification calibration (rAULC = –0.627, UCE = 0.417), suggesting that DEC improves OOD calibration only by sacrificing its ability to reflect in-distribution prediction errors. This contrast highlights CUPID's balanced and reliable uncertainty modeling across both ID and OOD scenarios.

### E.3 ABLATION AND HYPERPARAMETER EXPERIMENTS ON CUPID FOR CLASSIFICATION MODEL

Table 17: Performance of trunk block depth variants on CUPID for the classification task on HAM10000 dataset. Best-performing results for each metric are highlighted in bold.

| Method | | Aleatoric | | | Epistemic | | |
| | | AUC (↑) | AURC (↓) | Spearman (↑) | AUC (↑) | AURC (↓) | Spearman (↑) |
|---|---|---|---|---|---|---|---|
| **Block** | 12 | 0.725 ± 0.028 | **0.098 ± 0.010** | 0.614 ± 0.027 | 0.890 ± 0.003 | **0.052 ± 0.001** | **0.889 ± 0.004** |
| | 14 | 0.725 ± 0.013 | 0.107 ± 0.003 | 0.604 ± 0.035 | **0.902 ± 0.004** | 0.055 ± 0.003 | 0.886 ± 0.001 |
| | 16 | **0.749 ± 0.011** | 0.103 ± 0.006 | **0.675 ± 0.021** | 0.901 ± 0.004 | 0.058 ± 0.001 | 0.888 ± 0.005 |
| | 18 | 0.735 ± 0.024 | 0.100 ± 0.010 | 0.628 ± 0.049 | 0.895 ± 0.004 | 0.054 ± 0.002 | 0.886 ± 0.001 |

Table 18: Performance of loss function on CUPID for the classification task on HAM10000 dataset. Best-performing results for each metric are highlighted in bold.

| Loss | | Aleatoric | | | Epistemic | | |
| | AUC (↑) | AURC (↓) | Spearman (↑) | AUC (↑) | AURC (↓) | Spearman (↑) |
|---|---|---|---|---|---|---|---|
| Max | **0.749 ± 0.011** | 0.103 ± 0.006 | **0.675 ± 0.021** | **0.901 ± 0.004** | 0.058 ± 0.001 | **0.888 ± 0.005** |
| No max | 0.748 ± 0.005 | **0.098 ± 0.006** | 0.671 ± 0.025 | 0.898 ± 0.002 | **0.056 ± 0.003** | **0.888 ± 0.002** |

**Effect of Trunk Block Depth** Table 17 presents the impact of varying the number of Trunk Blocks in the Feature Extractor module of CUPID on HAM10000 dataset. CUPID is placed after the 2nd stage of residual blocks. For aleatoric uncertainty estimation, CUPID shows sensitivity to the depth of the trunk. The best performance is achieved when using 16 blocks, yielding the highest AUC of 0.749 and the highest Spearman correlation of 0.675. In contrast, the performance of epistemic uncertainty estimation is relatively stable across different depths, with only minor fluctuations in

AUC and AURC. This suggests that the epistemic branch reaches its representational capacity with fewer blocks, while aleatoric estimation benefits more from deeper feature extraction.

**Effect of Differential Feature Loss** We conduct an ablation study to evaluate the contribution of the differential feature loss term $-\|\mathbf{m_l} - \mathbf{m_l'}\|_1$, which enforces the difference between the intermediate feature $\mathbf{m_l}$ and the CUPID-generated reconstruction $\mathbf{m_l'}$. Table 18 summarizes the results for the misclassification detection task on the HAM10000 dataset. Including the differential feature loss slightly improves aleatoric performance in terms of AUC (from 0.748 to 0.749) and Spearman correlation (from 0.671 to 0.675), while maintaining comparable epistemic uncertainty performance.

For the out-of-distribution (OOD) detection task, the impact of the differential feature loss is more pronounced, particularly for epistemic uncertainty. As shown in Table 4, adding the loss substantially improves the performance of CUPID Epistemic on the PAPILA dataset, with AUC increasing from 0.839 to 0.877 and AUPR from 0.790 to 0.854. This indicates that the differential feature loss strengthens CUPID's sensitivity to distributional shifts. A similar trend is observed across the ACRIMA and CIFAR10 datasets, reinforcing the importance of this component for robust epistemic uncertainty estimation in OOD scenarios.

**Effect of $\lambda_1$ choice** The results in Table 19 and 20 show that CUPID is generally robust to the choice of $\lambda_1$. For misclassification detection, performance remains stable across all tested values. For OOD detection, $\lambda_1 = 0.01$ **consistently provides the best balance between stability and accuracy**, and is therefore used as the default throughout the paper.

Table 19: Performance of $\lambda_1$ choice on CUPID for the classification task on GLV2 dataset.

| Method | | Aleatoric | | | Epistemic | | |
|---|---|---|---|---|---|---|---|
| | | AUC (↑) | AURC (↓) | Spearman (↑) | AUC (↑) | AURC (↓) | Spearman (↑) |
| | 0.02 | 0.868 ± 0.001 | 0.018 ± 0.000 | 0.929 ± 0.019 | 0.743 ± 0.044 | 0.045 ± 0.013 | 0.676 ± 0.021 |
| $\lambda_1$ | 0.01 | **0.870 ± 0.002** | **0.018 ± 0.001** | **0.941 ± 0.004** | **0.769 ± 0.015** | **0.034 ± 0.002** | **0.701 ± 0.051** |
| | 0.001 | 0.868 ± 0.005 | 0.018 ± 0.001 | 0.936 ± 0.010 | 0.754 ± 0.014 | 0.042 ± 0.005 | 0.674 ± 0.031 |

Table 20: Performance of $\lambda_1$ choice on CUPID for the OOD task.

| Method and $\lambda_1$ | | PAPILA | | ACRIMA | | CIFAR10 | |
|---|---|---|---|---|---|---|---|
| | | AUC(↑) | AUPR(↑) | AUC(↑) | AUPR(↑) | AUC(↑) | AUPR(↑) |
| **Alea.** | 0.02 | 0.457 ± 0.087 | 0.362 ± 0.043 | 0.727 ± 0.124 | 0.651 ± 0.149 | 0.973 ± 0.017 | 0.996 ± 0.003 |
| | 0.01 | 0.379 ± 0.027 | 0.333 ± 0.007 | 0.717 ± 0.029 | 0.661 ± 0.027 | 0.983 ± 0.005 | 0.998 ± 0.001 |
| | 0.001 | 0.389 ± 0.030 | 0.342 ± 0.016 | 0.718 ± 0.046 | 0.669 ± 0.062 | **0.984 ± 0.006** | **0.999 ± 0.001** |
| **Epis.** | 0.02 | 0.802 ± 0.039 | 0.743 ± 0.058 | 0.904 ± 0.058 | 0.921 ± 0.046 | 0.528 ± 0.293 | 0.923 ± 0.063 |
| | 0.01 | **0.877 ± 0.032** | **0.854 ± 0.027** | **0.978 ± 0.010** | **0.984 ± 0.007** | 0.898 ± 0.054 | 0.991 ± 0.005 |
| | 0.001 | 0.836 ± 0.038 | 0.796 ± 0.051 | 0.975 ± 0.003 | 0.980 ± 0.002 | 0.896 ± 0.052 | 0.991 ± 0.005 |

### E.4 ABLATION AND HYPERPARAMETER EXPERIMENTS ON CUPID FOR SUPER-RESOLUTION

Table 21: Performance of trunk block depth variants on CUPID for the super-resolution task. Best-performing results for each metric are highlighted in bold.

| Trunk Num | | Set5 | | | Set14 | | |
|---|---|---|---|---|---|---|---|
| | | Pearson (↑) | AUSE (↓) | UCE (↓) | Pearson (↑) | AUSE (↓) | UCE (↓) |
| **Aleatoric** | 3 | 0.525 ± 0.002 | **0.010 ± 0.000** | **0.017 ± 0.007** | 0.527 ± 0.002 | **0.012 ± 0.000** | 0.049 ± 0.014 |
| | 4 | 0.528 ± 0.006 | **0.010 ± 0.000** | 0.045 ± 0.018 | 0.527 ± 0.002 | **0.012 ± 0.000** | 0.049 ± 0.005 |
| | 5 | 0.527 ± 0.007 | **0.010 ± 0.000** | 0.041 ± 0.019 | **0.527 ± 0.001** | **0.012 ± 0.000** | 0.047 ± 0.015 |
| | 6 | **0.528 ± 0.003** | **0.010 ± 0.000** | 0.045 ± 0.020 | 0.524 ± 0.003 | **0.012 ± 0.000** | **0.039 ± 0.006** |
| **Epistemic** | 3 | 0.416 ± 0.004 | 0.020 ± 0.000 | **0.254 ± 0.024** | 0.428 ± 0.005 | 0.020 ± 0.000 | 0.217 ± 0.003 |
| | 4 | 0.416 ± 0.005 | 0.018 ± 0.001 | 0.266 ± 0.007 | 0.449 ± 0.005 | **0.019 ± 0.000** | 0.226 ± 0.003 |
| | 5 | 0.420 ± 0.005 | **0.018 ± 0.000** | 0.256 ± 0.012 | 0.455 ± 0.004 | **0.019 ± 0.000** | 0.217 ± 0.011 |
| | 6 | **0.421 ± 0.005** | 0.018 ± 0.001 | 0.257 ± 0.014 | **0.459 ± 0.008** | **0.019 ± 0.000** | **0.211 ± 0.006** |

**Effect of Trunk Block Number**    Table 21 reports the performance of CUPID when varying the number of Trunk Blocks in the Feature Extractor on the Set5 and Set14 datasets. Overall, both aleatoric and epistemic uncertainty estimations remain relatively stable across different block depths. For aleatoric uncertainty, increasing the trunk number from 3 to 4 leads to a consistent improvement in Pearson correlation. While using 6 blocks achieves the best UCE on Set14 (0.039), the gains beyond 4 blocks are marginal. For epistemic uncertainty, deeper trunk configurations (especially 6 blocks) slightly improve the Pearson correlation, particularly on Set14 (from 0.428 to 0.459), and reduce UCE values modestly. These results suggest that while increasing the trunk depth can lead to minor performance gains, the CUPID framework remains robust even with fewer blocks.

**Effect of Perceptual Loss**    ESRGAN incorporates a perceptual loss derived from high-level feature representations extracted by a pre-trained VGG network, encouraging the model to generate outputs that are perceptually closer to the ground truth rather than strictly minimizing pixel-wise differences. Motivated by this, we investigate the effect of incorporating a perceptual loss into the CUPID training process.

Specifically, we augment the original CUPID loss function with a perceptual term, resulting in the following objective:

$$\mathcal{L}_{\text{CUPID}} = \mathcal{L}_{\text{Epis}} + \lambda_2 \mathcal{L}_{\text{Alea}} + \mathcal{L}_{\text{Lpips}}, \tag{38}$$

where $\mathcal{L}_{\text{Lpips}}$ denotes the perceptual loss computed using the LPIPS metric.

Table 22 presents the evaluation results on Set5 and Set14. While adding the perceptual loss decreases the Pearson correlation for both aleatoric and epistemic uncertainty (from 0.528 to 0.512 on Set5 for aleatoric), the calibration metrics AUSE and UCE remain largely unchanged or are slightly degraded. These findings suggest that although the perceptual loss enhances visual fidelity, it may introduce noise into the uncertainty estimation process, potentially due to the less pixel-aligned nature of perceptual features. As a result, we chose not to use perceptual loss.

Table 22: Performance of $\lambda_1$ choice on CUPID for the super-resolution task. Best-performing results for each metric are highlighted in bold.

| Loss | | Set5 | | | Set14 | | |
|---|---|---|---|---|---|---|---|
| | | **Pearson ($\uparrow$)** | **AUSE ($\downarrow$)** | **UCE ($\downarrow$)** | **Pearson ($\uparrow$)** | **AUSE ($\downarrow$)** | **UCE ($\downarrow$)** |
| **Aleatoric** | Raw | **0.528 ± 0.006** | **0.010 ± 0.000** | 0.045 ± 0.018 | **0.527 ± 0.002** | **0.012 ± 0.000** | **0.049 ± 0.005** |
| | Add Lipis | 0.512 ± 0.006 | 0.010 ± 0.001 | **0.031 ± 0.010** | 0.518 ± 0.009 | 0.013 ± 0.001 | 0.051 ± 0.015 |
| **Epistemic** | Raw | **0.416 ± 0.005** | **0.018 ± 0.001** | 0.266 ± 0.007 | **0.449 ± 0.005** | **0.019 ± 0.000** | **0.226 ± 0.003** |
| | Add Lipis | 0.389 ± 0.007 | 0.019 ± 0.000 | **0.253 ± 0.035** | 0.399 ± 0.003 | 0.021 ± 0.000 | 0.230 ± 0.016 |

**Effect of $\lambda_1$ choice**    As shown in Table 23, for super-resolution, the Pearson, AUSE, and UCE curves are **nearly identical** across the full range of $\lambda_1$ values, and $\lambda_1 = 0.001$ serves as a reliable default.

Table 23: Performance of $\lambda_1$ choice on CUPID for the super-resolution task.

| $\lambda_1$ | | Set5 | | | Set14 | | |
|---|---|---|---|---|---|---|---|
| | | **Pearson ($\uparrow$)** | **AUSE ($\downarrow$)** | **UCE ($\downarrow$)** | **Pearson ($\uparrow$)** | **AUSE ($\downarrow$)** | **UCE ($\downarrow$)** |
| **Aleatoric** | 0.001 | 0.527 ± 0.002 | **0.010 ± 0.000** | **0.034 ± 0.014** | 0.528 ± 0.001 | **0.012 ± 0.000** | 0.064 ± 0.016 |
| | 0.0001 | 0.528 ± 0.005 | **0.010 ± 0.000** | 0.045 ± 0.018 | 0.527 ± 0.002 | **0.012 ± 0.000** | **0.049 ± 0.005** |
| | 0.00001 | **0.529 ± 0.002** | **0.010 ± 0.000** | 0.037 ± 0.017 | **0.529 ± 0.002** | **0.012 ± 0.000** | 0.076 ± 0.011 |
| **Epistemic** | 0.001 | **0.423 ± 0.004** | **0.018 ± 0.000** | 0.264 ± 0.012 | **0.455 ± 0.010** | **0.018 ± 0.000** | 0.205 ± 0.009 |
| | 0.0001 | 0.416 ± 0.005 | **0.018 ± 0.001** | 0.266 ± 0.007 | 0.449 ± 0.005 | 0.019 ± 0.000 | 0.226 ± 0.003 |
| | 0.00001 | 0.421 ± 0.004 | **0.018 ± 0.001** | **0.262 ± 0.014** | **0.455 ± 0.011** | 0.019 ± 0.000 | **0.205 ± 0.005** |

### E.5    EFFECT OF ERROR MAP SELECTION

In regression-based tasks such as super-resolution, uncertainty metrics like Spearman correlation, AUSE, and UCE require comparison with an error map. We conduct experiments to evaluate how the choice of error map—$L_1$ or $L_2$—affects the alignment between the CUPID uncertainty map and the ground-truth error.

Table 25 presents the results for AUSE and UCE. We observe that using the $L_2$ error map generally leads to lower AUSE values, indicating better alignment with the overall uncertainty ranking. Conversely, the $L_1$ error map yields lower UCE scores, suggesting improved calibration of uncertainty magnitude. This reflects the distinct sensitivities of these metrics: AUSE focuses on ranking quality, while UCE measures calibration.

In addition, Table 24 shows the Spearman and Pearson correlations between the uncertainty maps and the $L_1/L_2$ error maps. Across both Set5 and Set14 datasets, $L_1$-based correlations are consistently higher than those computed with $L_2$, especially for Pearson correlation. Furthermore, Spearman values tend to exceed Pearson values, highlighting that CUPID uncertainty is more consistent with the rank ordering rather than the exact error values. These findings suggest that $L_1$ error maps may be more appropriate for evaluating uncertainty estimates in super-resolution tasks, particularly when ranking-based metrics are emphasized.

Table 24: Effect of error map selection on Pearson and Spearman metrics.

| Uncertainty | Error | Set5 | | Set14 | |
|---|---|---|---|---|---|
| | | Spearman ($\uparrow$) | Pearson ($\uparrow$) | Spearman ($\uparrow$) | Pearson ($\uparrow$) |
| Aleatoric | $L_1$ | $0.643 \pm 0.002$ | $\mathbf{0.560 \pm 0.001}$ | $0.607 \pm 0.002$ | $\mathbf{0.569 \pm 0.001}$ |
| | $L_2$ | $\mathbf{0.647 \pm 0.002}$ | $0.487 \pm 0.003$ | $\mathbf{0.611 \pm 0.002}$ | $0.514 \pm 0.004$ |
| Epistemic | $L_1$ | $\mathbf{0.229 \pm 0.007}$ | $\mathbf{0.258 \pm 0.005}$ | $\mathbf{0.312 \pm 0.006}$ | $\mathbf{0.327 \pm 0.005}$ |
| | $L_2$ | $0.229 \pm 0.008$ | $0.209 \pm 0.005$ | $\mathbf{0.312 \pm 0.006}$ | $0.254 \pm 0.003$ |

Table 25: Effect of error map selection on AUSE and UCE metrics.

| Uncertainty | Error | Set5 | | Set14 | |
|---|---|---|---|---|---|
| | | AUSE ($\downarrow$) | UCE ($\downarrow$) | AUSE ($\downarrow$) | UCE ($\downarrow$) |
| Aleatoric | $L_1$ | $0.009 \pm 0.000$ | $\mathbf{0.034 \pm 0.007}$ | $0.011 \pm 0.000$ | $\mathbf{0.017 \pm 0.008}$ |
| | $L_2$ | $\mathbf{0.002 \pm 0.000}$ | $0.358 \pm 0.004$ | $\mathbf{0.002 \pm 0.000}$ | $0.269 \pm 0.021$ |
| Epistemic | $L_1$ | $0.026 \pm 0.000$ | $\mathbf{0.320 \pm 0.004}$ | $0.023 \pm 0.000$ | $\mathbf{0.231 \pm 0.005}$ |
| | $L_2$ | $\mathbf{0.005 \pm 0.000}$ | $0.420 \pm 0.001$ | $\mathbf{0.004 \pm 0.000}$ | $0.411 \pm 0.001$ |

### E.6 MORE VISUAL RESULTS

Figure 8 presents the visual results of the proposed CUPID framework on single-image super-resolution. As illustrated in the second row, the uncertainty maps produced by CUPID closely align with the pixel-wise $L_1$ and $L_2$ error maps. This consistency suggests that CUPID accurately captures regions of high reconstruction error, reflecting its ability to localize uncertainty. Additionally, the difference between the original intermediate feature and the reconstructed feature generated by CUPID indicates that the Reconstruction Branch tends to enhance fine-grained image details, particularly in high-frequency regions such as edges and textures.

Figure 9 presents the visual results of the proposed CUPID framework and comparison methods.

### E.7 COMPARISON OF RUNTIME ACROSS DIFFERENT UNCERTAINTY ESTIMATES.

Table 26 and Table 27 presents the runtime for training and prediction across all models evaluated in this study. The training time reflects the total duration required to train each model on the EyePACS training dataset, while the prediction time indicates the time taken to generate predictions for the EyePACS test dataset.

Table 26: Runtime comparison between training and testing phases in classification model uncertainty estimation.

| Operation | Training Time (s) | Testing Time (s) |
|---|---|---|
| CUPID | 4776.38 | 10.06 |
| MC Dropout | / | 49.57 |
| Rate-in | 2.85 | 48.46 |
| IGRUE | 5292.41 | 11.26 |
| PostNet | 49932.08 | 51.04 |
| BNN | 54974.84 | 217.06 |
| DEC | 1615.58 | 6.46 |

Table 27: Runtime comparison between training and testing phases in regression model uncertainty estimation.

| Operation | Training Time (s) | Testing Time (s) |
|---|---|---|
| CUPID | 26511.74 | 1.02 |
| BayesCap | 81158.45 | 0.79 |
| in-rotate | / | 2.03 |
| in-noise | / | 2.86 |
| med-dropout | / | 2.28 |
| med-noise | / | 2.47 |

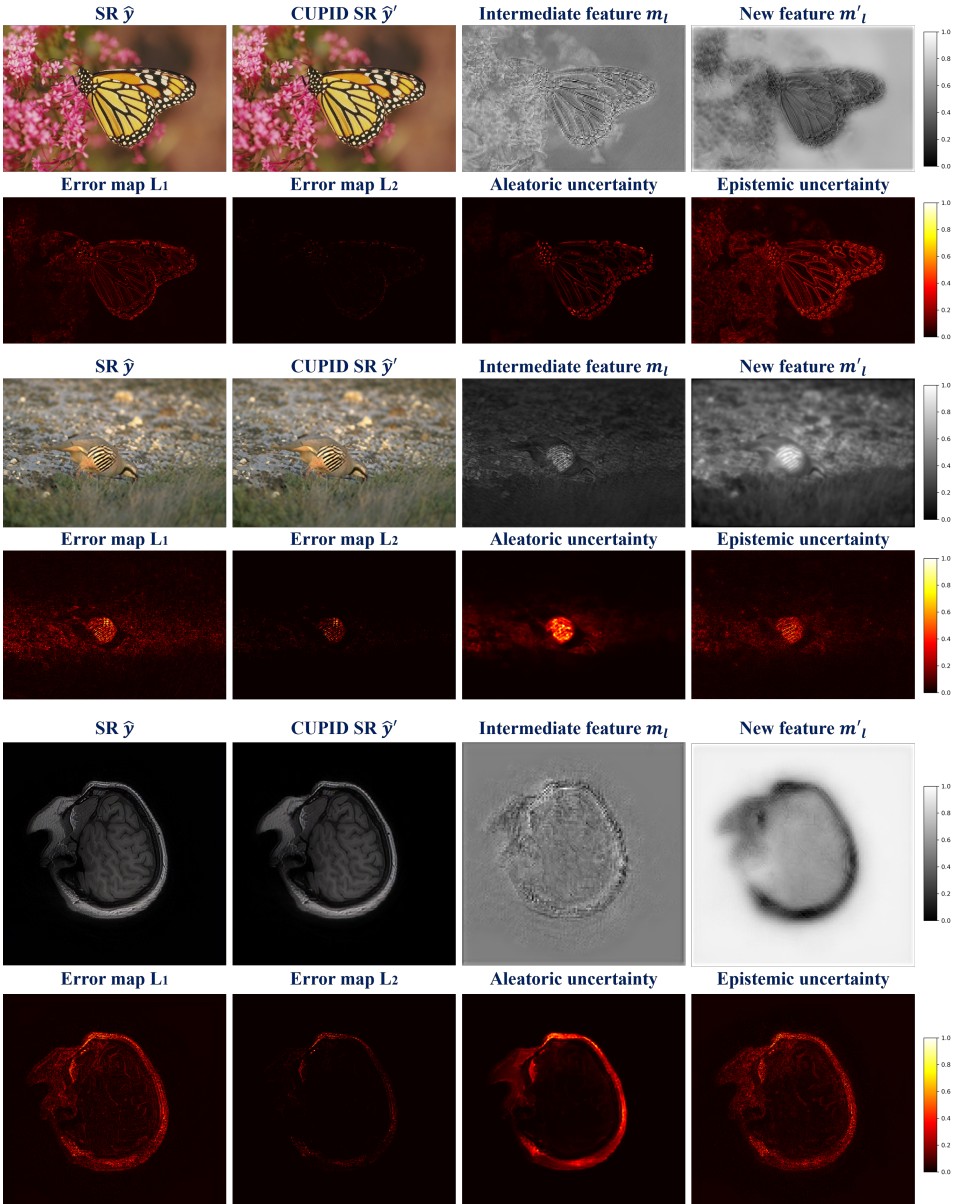

Figure 8: More visual results of the uncertainty feature map of CUPID towards super-resolution model.

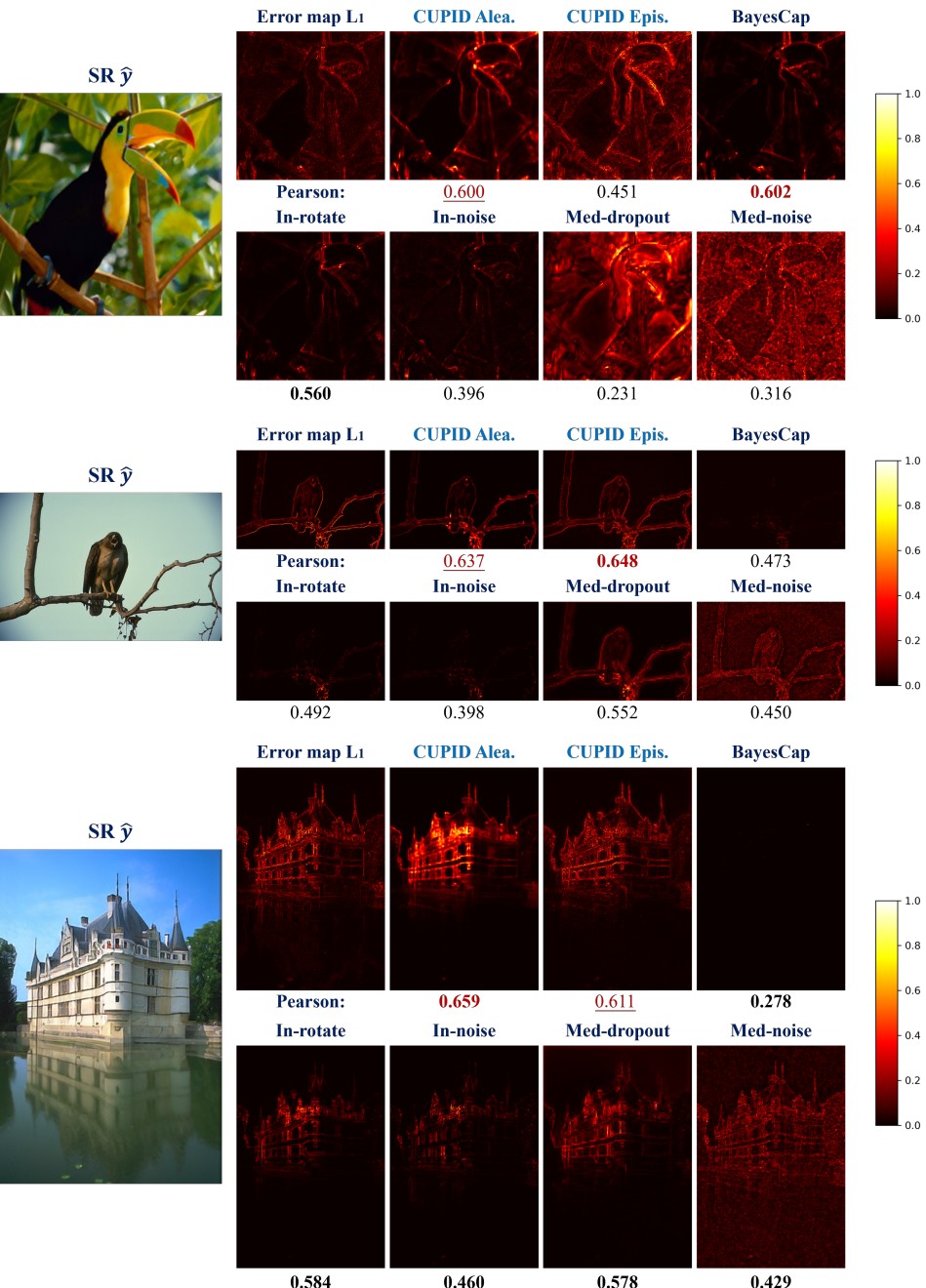

Figure 9: More visual results of the uncertainty feature map of all the methods.

