# OpenReview forum: "CUPID: A Plug-in Framework for Joint Aleatoric and Epistemic Uncertainty Estimation with a Single Model"
_ICLR.cc/2026/Conference — ICLR 2026 Poster_

### Official Review · Reviewer_7Juj · 2025-10-28

**Soundness:** 4
**Presentation:** 4
**Contribution:** 3
**Rating:** 8
**Confidence:** 4

**Summary:**

This paper introduces Comprehensive Uncertainty Plug-in estImation moDel (CUPID), an uncertainty estimation method that quantifies both epistemic uncertainty (EU) and aleatoric uncertainty (AU) without requiring retraining of the base model. The method introduces a CUPID module, which operates on top of intermediate network features to estimate uncertainty. This module learns to capture AU through a log-likelihood objective and EU through a loss which encourages maximal feature differences and minimal reconstruction differences. Experiments show performance of CUPID on downstream tasks like misclassification detection, out-of-distribution (OOD) detection, and regression.

**Strengths:**

1. Novelty of the proposed method. The proposed objectives and CUPID module are novel ideas that seem quite robust and well-performant for downstream applications. The method also has the benefit of not requiring retraining of the base model, reducing computational overhead.
2. Clarity. The presentation of the paper is well organized and discussion are clear / easy to follow.
3. Experimental results. The experiments cover a good number of baselines and applications, giving a good benchmark of how the method is expected to perform in real-world scenarios.

**Weaknesses:**

1. Computational analysis. While the method doesn't require retraining of the base model, some metrics (e.g., runtime, memory consumption) of CUPID should be provided to give an idea of the expected computational overhead.
2. For the misclassification experiment in Section 4.1, it would be good to find some empirical evidence that GLV2 is AU dominant and HAM1000 is EU dominant (i.e., not just by looking at other baseline results, but by examining at the data itself).

**Questions:**

1. I'm a little unclear on the intuition behind the EU loss function in Section 3.3. Why do we want the intermediate features to be maximally different, but the outputs to be minimally different?
2. What are the recommended strategies for choosing $\lambda_1$ (e.g., is there a fixed value that tends to work well or does this need to be carefully tuned)?

---

> ### Author Response · Authors · 2025-11-22
> **Reply to W1**
>
> We sincerely thank the reviewer for the very positive and constructive feedback. We appreciate the recognition of CUPID's novelty, clarity, and practical value. In the revised version, we included computational overhead analysis, data-driven evidence for AU/EU dominance, and additional clarifications of the EU objective and hyperparameter choices.
>
> **1. W1:**  *Computational analysis. While the method doesn't require retraining of the base model, some metrics (e.g., runtime, memory consumption) of CUPID should be provided to give an idea of the expected computational overhead.*
>
> **R1.** Thank you for pointing this out. We have added a detailed runtime comparison in the revised appendix as Table 27 and Table 28. CUPID introduces only a lightweight plug-in module and requires exactly **two forward passes** (original and perturbed), without retraining or modifying the base model. Consequently, its inference cost is substantially lower than methods that rely on repeated stochastic sampling (MC Dropout, in-noise) or multiple full model evaluations (PostNet, BNN). Empirically, CUPID achieves competitive or lower testing time while maintaining strong performance.
>
> **Table 27: Runtime comparison between training and testing phases for uncertainty estimation in classification.**
>
> | **Operation**  | **Training Time (s)** | **Testing Time (s)** |
> | -------------- | --------------------- | -------------------- |
> | **CUPID**      | 4776.38               | 10.06                |
> | **MC Dropout** | /                     | 49.57                |
> | **Rate-in**    | 2.85                  | 48.46                |
> | **IGRUE**      | 5292.41               | 11.26                |
> | **PostNet**    | 49932.08              | 51.04                |
> | **BNN**        | 54974.84              | 217.06               |
> | **DEC**        | 1615.58               | 6.46                 |
>
> **Table 28: Runtime comparison between training and testing phases for uncertainty estimation in regression.**
>
> | **Operation**   | **Training Time (s)** | **Testing Time (s)** |
> | --------------- | --------------------- | -------------------- |
> | **CUPID**       | 26511.74              | 1.02                 |
> | **BayesCap**    | 81158.45              | 0.79                 |
> | **in-rotate**   | /                     | 2.03                 |
> | **in-noise**    | /                     | 2.86                 |
> | **med-dropout** | /                     | 2.28                 |
> | **med-noise**   | /                     | 2.47                 |

---

> > ### Author Response · Authors · 2025-11-22
> > **Reply to W2**
> >
> > **2. W2:**  *For the misclassification experiment in Section 4.1, it would be good to find some empirical evidence that GLV2 is AU dominant and HAM1000 is EU dominant (i.e., not just by looking at other baseline results, but by examining at the data itself).*
> >
> > **R2.** We appreciate the reviewer’s suggestion and provide direct dataset-level evidence supporting the AU–EU differences between GLV2 and HAM10000.
> >
> > **Aleatoric Uncertainty (AU).** We quantify acquisition-related randomness using dataset-level pixel variance (“Noise”) and signal-to-noise ratio (SNR)} [1], computed from the first- and second-order moments of all images. Higher Noise and lower SNR indicate stronger imaging variability, which is a typical driver of aleatoric uncertainty. As shown in Table 10, GLV2 exhibits higher Noise (0.024 vs. 0.022) and markedly lower SNR (5.44,dB vs. 12.78,dB) than HAM10000, confirming that GLV2 contains more acquisition-induced variability and is therefore more AU-dominant.
> >
> > **Epistemic Uncertainty (EU).**  EU arises from limited or uneven data coverage [2]. HAM10000 contains 7 diagnostic classes with substantial imbalance (142–6705 samples), forming a heterogeneous and sparsely supported feature space. In contrast, GLV2 consists of only 2 well-balanced classes. This structural difference implies that models trained on HAM10000 must learn more complex and uneven decision boundaries, resulting in higher epistemic uncertainty, while GLV2’s simpler and balanced distribution yields lower EU. These dataset characteristics (Table 10) directly explain the observed AU/EU behavior in the main experiments. We also provide visual examples in the appendix Figure 5.
> >
> > **Table 10: Dataset characteristics indicating different sources of uncertainty.**
> >
> > | **Dataset** | **Noise ↑** | **SNR ↓**  | **Classes** | **Class Balanced** |
> > | ----------- | --------- | -------- | ----------- | ------------------ |
> > | GLV2        | 0.024     | 5.44 dB  | 2           | True               |
> > | HAM10000    | 0.022     | 12.78 dB | 7          | False              |

---

> ### Author Response · Authors · 2025-11-22
> **Reply to Q1**
>
> **3. Q1:** *I'm a little unclear on the intuition behind the EU loss function in Section 3.3. Why do we want the intermediate features to be maximally different, but the outputs to be minimally different?*
>
> **R3.** We thank the reviewer for this important conceptual question. CUPID’s epistemic uncertainty objective is based on the principle that a model is confident only when its prediction is locally stable: the output should remain unchanged even when the internal representation is perturbed. Conversely, if even a small feature shift leads to an output change, the model is operating in a region of high epistemic uncertainty.
>
> **Why maximize feature deviation.** Maximizing $\||\mathbf{m}'-\mathbf{m}\||$ prevents CUPID from degenerating into an autoencoder that simply reproduces the original features, and instead forces it to learn how the prediction behaves when the representation moves away from the training manifold.
>
> **Formal intuition.** A first-order expansion gives
>
> $U\_{\text{epis}}(\mathbf{x})
> \approx
> \underbrace{
> ||
> \nabla\_{\mathbf{m}\_{l,n}} F\_l(\mathbf{m}\_{l,n})
> \cdot
> (\mathbf{m}'\_{l,n}-\mathbf{m}\_{l,n})
> ||\_1
> }\_{%
> \text{Sensitivity}
> \times
> \text{Deviation}
> }$
>
> Thus, CUPID captures epistemic uncertainty through two complementary factors: the model’s local sensitivity (a classical EU signal [3]) and the size of the feature-space region over which the prediction remains stable. When an input is out-of-distribution, the learned perturbation naturally produces larger and less stable deviations, revealing abnormal epistemic behavior. In the revised manuscript, we validated the importance of the feature-deviation term through an ablation study (Section 4.4) shown in Table 4. Removing the maximization component significantly weakens epistemic discrimination: on PAPILA OOD detection, AUC drops from 0.877 to 0.839 and AUPR from 0.854 to 0.790.
>
> In summary, maximizing feature deviation explores unsupported feature directions, while minimizing output change isolates true epistemic sensitivity. Both components are necessary for CUPID to capture meaningful epistemic uncertainty.
>
> **Table 4: Performance of differential feature loss on OOD task. "No max" means remove $-\||\mathbf{m}\_{l,n} - \mathbf{m}\_{l,n}'\||\_1$ in the loss function. Best-performing results for each metric are highlighted in bold.**
>
> | **Method** | **Type** | **PAPILA AUC ↑**  | **PAPILA AUPR ↑** | **ACRIMA AUC ↑**  | **ACRIMA AUPR ↑** | **CIFAR-10 AUC ↑** | **CIFAR-10 AUPR ↑** |
> | ---------- | -------- | ----------------- | ----------------- | ----------------- | ----------------- | ------------------ | ------------------- |
> | Max        | Alea.    | 0.379 ± 0.027     | 0.333 ± 0.007     | 0.717 ± 0.029     | 0.661 ± 0.027     | 0.983 ± 0.005      | 0.998 ± 0.001       |
> | No max     | Alea.    | 0.389 ± 0.026     | 0.338 ± 0.009     | 0.739 ± 0.042     | 0.696 ± 0.055     | **0.988 ± 0.003**  | **0.999 ± 0.000**   |
> | Max        | Epis.    | **0.877 ± 0.032** | **0.854 ± 0.027** | **0.978 ± 0.010** | **0.984 ± 0.007** | 0.898 ± 0.054      | 0.991 ± 0.005       |
> | No max     | Epis.    | 0.839 ± 0.017     | 0.790 ± 0.054     | 0.977 ± 0.006     | 0.982 ± 0.005     | 0.875 ± 0.024      | 0.989 ± 0.002       |

---

> ### Author Response · Authors · 2025-11-22
> **Reply to Q2**
>
> **4. Q2:**  *What are the recommended strategies for choosing $\lambda\_1$ (e.g., is there a fixed value that tends to work well or does this need to be carefully tuned)?*
>
> **R4.** We conducted a sensitivity analysis across misclassification, OOD detection, and super-resolution tasks (added in revised Appendix as Table 17-19). The results show that CUPID is generally robust to the choice of $\lambda\_1$. For misclassification detection, performance remains stable across all tested values. For OOD detection, $\lambda\_1=0.01$ consistently provides the best balance between stability and accuracy, and is therefore used as the default throughout the paper. For super-resolution, the Pearson, AUSE, and UCE curves are nearly identical across the full range of $\lambda\_1$ values, and $\lambda\_1=0.001$ serves as a reliable default. Overall, the experiments demonstrate that $\lambda\_1$ only mildly affects OOD detection and has negligible impact on misclassification and regression tasks, confirming that CUPID does not require careful hyperparameter tuning.
>
> **Table 17: Performance of different λ₁ values for CUPID on the GLV2 classification task.**
>
> | $\lambda\_1$ | **Aleatoric AUC ↑** | **Aleatoric AURC ↓** | **Aleatoric Spearman ↑** | **Epistemic AUC ↑** | **Epistemic AURC ↓** | **Epistemic Spearman ↑** |
> | ------------ | ------------------- | -------------------- | ------------------------ | ------------------- | -------------------- | ------------------------ |
> | 0.02         | 0.868 ± 0.001       | 0.018 ± 0.000        | 0.929 ± 0.019            | 0.743 ± 0.044       | 0.045 ± 0.013        | 0.676 ± 0.021            |
> | 0.01         | **0.870 ± 0.002**   | **0.018 ± 0.001**    | **0.941 ± 0.004**        | **0.769 ± 0.015**   | **0.034 ± 0.002**    | **0.701 ± 0.051**        |
> | 0.001        | 0.868 ± 0.005       | 0.018 ± 0.001        | 0.936 ± 0.010            | 0.754 ± 0.014       | 0.042 ± 0.005        | 0.674 ± 0.031            |
>
> **Table 18: Performance of different λ₁ values for CUPID on OOD detection (PAPILA, ACRIMA, CIFAR-10).**
>
> | **Method** | **$\lambda\_1$** | **PAPILA AUC ↑**  | **PAPILA AUPR ↑** | **ACRIMA AUC ↑**  | **ACRIMA AUPR ↑** | **CIFAR-10 AUC ↑** | **CIFAR-10 AUPR ↑** |
> | ---------- | ---------------- | ----------------- | ----------------- | ----------------- | ----------------- | ------------------ | ------------------- |
> | **Alea.**  | 0.02             | 0.457 ± 0.087     | 0.362 ± 0.043     | 0.727 ± 0.124     | 0.651 ± 0.149     | 0.973 ± 0.017      | 0.996 ± 0.003       |
> | **Alea.**  | 0.01             | 0.379 ± 0.027     | 0.333 ± 0.007     | 0.717 ± 0.029     | 0.661 ± 0.027     | 0.983 ± 0.005      | 0.998 ± 0.001       |
> | **Alea.**  | 0.001            | 0.389 ± 0.030     | 0.342 ± 0.016     | 0.718 ± 0.046     | 0.669 ± 0.062     | **0.984 ± 0.006**  | **0.999 ± 0.001**   |
> | **Epis.**  | 0.02             | 0.802 ± 0.039     | 0.743 ± 0.058     | 0.904 ± 0.058     | 0.921 ± 0.046     | 0.528 ± 0.293      | 0.923 ± 0.063       |
> | **Epis.**  | 0.01             | **0.877 ± 0.032** | **0.854 ± 0.027** | **0.978 ± 0.010** | **0.984 ± 0.007** | 0.898 ± 0.054      | 0.991 ± 0.005       |
> | **Epis.**  | 0.001            | 0.836 ± 0.038     | 0.796 ± 0.051     | 0.975 ± 0.003     | 0.980 ± 0.002     | 0.896 ± 0.052      | 0.991 ± 0.005       |
>
> **Table: Performance of different λ₁ values for CUPID on the super-resolution task (Set5, Set14).**
>
> | **Method**    | **λ₁**  | **Set5 Pearson ↑** | **Set5 AUSE ↓**   | **Set5 UCE ↓**    | **Set14 Pearson ↑** | **Set14 AUSE ↓**  | **Set14 UCE ↓**   |
> | ------------- | ------- | ------------------ | ----------------- | ----------------- | ------------------- | ----------------- | ----------------- |
> | **Alea.** | 0.001   | 0.527 ± 0.002      | **0.010 ± 0.000** | **0.034 ± 0.014** | 0.528 ± 0.001       | **0.012 ± 0.000** | 0.064 ± 0.016     |
> | **Alea.** | 0.0001  | 0.528 ± 0.006      | **0.010 ± 0.000** | 0.045 ± 0.018     | 0.527 ± 0.002       | **0.012 ± 0.000** | **0.049 ± 0.005** |
> | **Alea.** | 0.00001 | **0.529 ± 0.002**  | **0.010 ± 0.000** | 0.037 ± 0.017     | **0.529 ± 0.002**   | **0.012 ± 0.000** | 0.076 ± 0.011     |
> | **Epis.** | 0.001   | **0.423 ± 0.004**  | **0.018 ± 0.000** | 0.264 ± 0.012     | **0.455 ± 0.010**   | **0.018 ± 0.000** | **0.205 ± 0.009** |
> | **Epis.** | 0.0001  | 0.416 ± 0.005      | **0.018 ± 0.001** | 0.266 ± 0.007     | 0.449 ± 0.005       | 0.019 ± 0.000     | 0.226 ± 0.003     |
> | **Epis.** | 0.00001 | 0.421 ± 0.004      | **0.018 ± 0.001** | **0.262 ± 0.014** | **0.455 ± 0.011**   | 0.019 ± 0.000     | **0.205 ± 0.005** |
>
> We thank the reviewer again for the valuable insights. All suggestions will improve clarity and completeness of the paper, and we will incorporate the above additions in the revised supplementary material.

---

> > ### Author Response · Authors · 2025-11-22
> > **Reference**
> >
> > **Reference**
> >
> > [1] Subham Sahoo, Huai Wang, and Frede Blaabjerg. Uncertainty-aware artificial intelligence for gear fault diagnosis in motor drives. In 2025 IEEE Applied Power Electronics Conference and Exposition (APEC), pp. 912–918. IEEE, 2025.
> >
> > [2] Ke Zou, Zhihao Chen, Xuedong Yuan, Xiaojing Shen, Meng Wang, and Huazhu Fu. A review of uncertainty estimation and its application in medical imaging. Meta-Radiology, pp. 100003, 2023.
> >
> > [3] Hanjing Wang and Qiang Ji. Epistemic uncertainty quantification for pre-trained neural networks. In Proceedings of the IEEE/CVF Conference on Computer Vision and Pattern Recognition, pp. 11052–11061, 2024.

---

### Official Review · Reviewer_D4Tj · 2025-10-30

**Soundness:** 2
**Presentation:** 2
**Contribution:** 2
**Rating:** 4
**Confidence:** 4

**Summary:**

The CUPID framework proposes a model-agnostic, plug-in method for estimating and decomposing aleatoric and epistemic uncertainties within deep neural networks. Instead of retraining or modifying the base model, CUPID attaches an auxiliary uncertainty branch to intermediate feature layers, allowing uncertainty estimation through a single forward pass. The aleatoric component is predicted via a regression branch that models data-dependent variance, while the epistemic component is derived from the feature reconstruction error of a decoder, both regularized under a Bayesian identity mapping to prevent mutual interference. This layer-wise design enables localized uncertainty interpretation and can be flexibly applied to pretrained networks across different tasks. Experimental results on classification, regression, and OOD detection demonstrate that CUPID achieves competitive or superior calibration and interpretability compared to existing post-hoc and evidential approaches, highlighting its practicality as a lightweight uncertainty quantification module.

**Strengths:**

The proposed CUPID framework offers a simple yet flexible plug-in design that can be applied to pretrained models without retraining. This practical modularity makes the method attractive for real-world applications where uncertainty estimation must be added post hoc. The joint decomposition of aleatoric and epistemic uncertainties provides a unified view of model and data uncertainty, and the layer-wise formulation enables interpretable analysis of which network components contribute most to uncertainty. These design choices improve usability and diagnostic insight with minimal computational overhead.

**Weaknesses:**

**Methods**

The proposed methods for estimating aleatoric and epistemic uncertainty are conceptually straightforward and largely aligned with prior works. The aleatoric branch essentially performs feature-based variance regression, similar to heteroscedastic uncertainty modeling in Kendall & Gal (2017) and Evidential Deep Learning (Sensoy et al., 2018). The epistemic uncertainty is derived from feature reconstruction errors, an idea previously explored in autoencoder-based OOD detection (An & Cho, 2015) and variational approaches. Therefore, the underlying mechanisms are not fundamentally novel. The contribution of CUPID mainly lies in its plug-in, layer-wise formulation and the joint decomposition enabled by the Bayesian identity mapping, which improves modularity and interpretability but does not constitute a substantial algorithmic innovation.

References
An, J. & Cho, S. (2015). Variational Autoencoder based Anomaly Detection using Reconstruction Probability. — Workshop on Machine Learning for Signal Processing (MLSP), IEEE.


**Experiments**

Although Section 3.4 describes a unified objective for jointly training the aleatoric and epistemic branches, the experiments appear to use separately trained versions (CUPID-A and CUPID-E) rather than a fully joint model. The paper would benefit from clarifying whether the joint training was actually performed, and if so, providing explicit results comparing the jointly trained version with the individual branches. Otherwise, the claimed joint decomposition may remain only conceptual rather than empirically validated.

**Questions:**

Q1.
In Tables 1, 2, and 3, I could not find the results of the unified model that jointly trains the aleatoric and epistemic branches. Moreover, the performance trends between the aleatoric and epistemic versions appear quite contradictory, suggesting that joint training might degrade performance. This inconsistency limits the practical benefit of the proposed method, as it becomes cumbersome to decide which type of uncertainty is more suitable for a given task.

Q2.
Using feature reconstruction as a proxy for epistemic uncertainty is not new. What distinguishes CUPID’s reconstruction-based approach from prior autoencoder-based OOD detection or evidential methods? Furthermore, is there any theoretical justification for interpreting reconstruction error specifically as epistemic uncertainty rather than as a general measure of representation distance?

Q3.
The aleatoric branch directly regresses variance from intermediate feature activations. How is this branch trained in classification settings where ground-truth noise levels are unavailable? Are the predicted variances calibrated, normalized, or otherwise constrained to ensure meaningful uncertainty estimates?

---

> ### Author Response · Authors · 2025-11-22
> **Reply to W1**
>
> We thank the reviewer for the constructive and thoughtful feedback. We appreciate the recognition of CUPID’s practicality, interpretability, and modularity, and we are grateful for the suggestions that help us improve clarity regarding joint training, the distinction from prior reconstruction-based methods, and the aleatoric learning mechanism. Below, we respond to each concern point-by-point, clarify misunderstandings, and provide additional experimental and theoretical evidence that strengthens our contributions.
>
> **1. W1 (Methods):** *Aleatoric regression resembles heteroscedastic modeling; epistemic uncertainty resembles reconstruction-based OOD detection.*
>
> **R1:** We appreciate the reviewer’s concern and clarify that CUPID’s contribution does not lie in redefining heteroscedastic regression or reconstruction only, but in introducing the **first plug-in module capable of jointly estimating aleatoric and epistemic uncertainty inside the intermediate layers of a frozen network**. Existing approaches either (1) require full retraining of the base model (BNN, EDL), (2) estimate only epistemic uncertainty (RUE, gradient-based perturbation methods), (3) estimate only aleatoric uncertainty (heteroscedastic regression). None offer CUPID’s combination of joint decomposition, layer-wise placement, and two-forward-pass inference without modifying the pretrained architecture.
>
> CUPID’s aleatoric branch indeed builds upon the heteroscedastic formulation, but its effectiveness is strengthened by joint training with the epistemic branch (see R4). More importantly, CUPID’s epistemic branch is fundamentally different from reconstruction-based OOD detection: it does **not minimize reconstruction error but instead optimizes structured feature perturbations under prediction invariance**, a mechanism not present in prior autoencoder or reconstruction-based methods. We elaborate on this distinction in R2.

---

> > ### Author Response · Authors · 2025-11-22
> > **Reply to Q2**
> >
> > **2. Q2:** *What distinguishes CUPID from autoencoder-based reconstruction methods? How is reconstruction error interpreted as epistemic uncertainty?*
> >
> > **R2.** We first clarify that CUPID does **not rely on reconstruction loss**. Although it outputs a perturbed intermediate feature, CUPID never minimizes $||\mathbf{m}'\_{l,n} - \mathbf{m}\_{l,n}||$ as in autoencoders. Instead, it learns perturbations by **maximizing** feature deviation while enforcing **prediction consistency**:
> >
> > $max\_{\mathbf{m}'\_{l,n}} ||\mathbf{m}'\_{l,n}-\mathbf{m}\_{l,n}||\_1
> > \qquad
> > \min\_{\mathbf{m}'\_{l,n}} ||\hat{\mathbf{y}}'\_n-\hat{\mathbf{y}}\_n||\_1.
> > $
> >
> > Here, $\mathbf{m}\_{l,n}$ denotes the intermediate feature of sample $n$ at layer $l$, $\mathbf{m}'\_{l,n}$ is the perturbed feature produced by CUPID, $\hat{\mathbf{y}}\_n = F\_l(\mathbf{m}\_{l,n})$ is the original model output, and $\hat{\mathbf{y}}'\_n = F\_l(\mathbf{m}'\_{l,n})$ is the output under perturbation. We also revised the Figure 2 to emphasize this mechanism. This mechanism differs fundamentally from prior VAE- or feature-reconstruction OOD detectors, which minimize reconstruction error and thus capture input regularity rather than model uncertainty [1].
> >
> > **CUPID’s epistemic uncertainty.** CUPID defines epistemic uncertainty as the output discrepancy under the learned, prediction-preserving perturbation:
> > $U\_{\text{epis}}(\mathbf{x}) := ||F\_l(\mathbf{m}\_{l,n}) - F\_l(\mathbf{m}'\_{l,n})||\_1.$
> > This aligns with the classical view that epistemic uncertainty reflects the model’s sensitivity to changes in parts of the representation space [2]. Unlike random perturbation methods, which require many forward passes, CUPID learns a structured perturbation direction once, enabling single-pass inference.
> >
> > **Why this corresponds to epistemic uncertainty.** A first-order expansion of $F\_l$ gives:
> >
> > $U\_{\text{epis}}(\mathbf{x})
> > \approx
> > \underbrace{
> > ||
> > \nabla\_{\mathbf{m}\_{l,n}} F\_l(\mathbf{m}\_{l,n})
> > \cdot
> > (\mathbf{m}'\_{l,n}-\mathbf{m}\_{l,n})
> > ||\_1
> > }\_{%
> > \text{Sensitivity}
> > \times
> > \text{Deviation}
> > }$
> >
> > Sensitivity captures how unstable the model is around $\mathbf{m}\_{l,n}$, a well-known indicator of epistemic uncertainty [3,4]. Deviation measures how far the representation can move while maintaining prediction consistency, reflecting the model’s confidence in that region of the feature space. CUPID therefore jointly captures (1) model sensitivity and (2) allowed deviation from the training manifold, providing a principled approximation to epistemic uncertainty. In-distribution misclassified samples often exhibit high sensitivity, while OOD samples induce abnormally large deviation. CUPID therefore provides a unified estimate of epistemic uncertainty that responds to both failure modes.
> >
> > We highlight this effect empirically: adding the differential feature loss significantly improves epistemic performance under distribution shift (PAPILA AUC from $0.839$ to $0.877$, AUPR from $0.790$ to $0.854$), showing that CUPID’s deviation-based formulation is effective for modeling epistemic uncertainty.
> >
> > **Table 4: Performance of differential feature loss on OOD task. "No max" means remove $-||\mathbf{m}\_{l,n} - \mathbf{m}\_{l,n}'||\_1$ in the loss function. Best-performing results for each metric are highlighted in bold.**
> >
> > | **Method** | **Type** | **PAPILA AUC ↑**  | **PAPILA AUPR ↑** | **ACRIMA AUC ↑**  | **ACRIMA AUPR ↑** | **CIFAR-10 AUC ↑** | **CIFAR-10 AUPR ↑** |
> > | ---------- | -------- | ----------------- | ----------------- | ----------------- | ----------------- | ------------------ | ------------------- |
> > | Max        | Alea.    | 0.379 ± 0.027     | 0.333 ± 0.007     | 0.717 ± 0.029     | 0.661 ± 0.027     | 0.983 ± 0.005      | 0.998 ± 0.001       |
> > | No max     | Alea.    | 0.389 ± 0.026     | 0.338 ± 0.009     | 0.739 ± 0.042     | 0.696 ± 0.055     | **0.988 ± 0.003**  | **0.999 ± 0.000**   |
> > | Max        | Epis.    | **0.877 ± 0.032** | **0.854 ± 0.027** | **0.978 ± 0.010** | **0.984 ± 0.007** | 0.898 ± 0.054      | 0.991 ± 0.005       |
> > | No max     | Epis.    | 0.839 ± 0.017     | 0.790 ± 0.054     | 0.977 ± 0.006     | 0.982 ± 0.005     | 0.875 ± 0.024      | 0.989 ± 0.002       |

---

> ### Author Response · Authors · 2025-11-22
> **Reply to W2 & Q1**
>
> **3. W2 (Experiments) and Q1:** *I cannot find results of the jointly trained model; aleatoric and epistemic trends look inconsistent.*
>
> **R3:** We appreciate the insightful comment. **Tables 1--3 already report the jointly trained CUPID model.** For clarity, we emphasize that *CUPID-Alea.* and *CUPID-Epis.* in the joint configuration are derived from the **same jointly trained model**; these names simply denote the aleatoric and epistemic outputs of CUPID’s two-branch structure.
>
> We have now explicitly clarified this in the Section 3.4 in revision:
>
> > Both the epistemic and aleatoric estimation are optimized simultaneously under this unified loss, ensuring that CUPID learns both uncertainty types within a single model.
>
> We also added a joint-vs-separate comparison table in the revised manuscipte in section 4.4, showing that the unified model performs better comparable to separate branches across all tasks. This confirms that joint training is stable and does not degrade performance.
>
> > **Joint vs. Separate Training.**  Both the reconstruction branch (epistemic) and the uncertainty branch (aleatoric) in CUPID are present and jointly optimized. To evaluate whether this two-branch architecture provides mutual benefit, we conduct an ablation study in which we remove one branch entirely and train the remaining branch in isolation. Specifically, (1) *Alea. separate* denotes a model where the epistemic branch is removed and only the aleatoric branch is trained, and (2) *Epis. separate* denotes a model where the aleatoric branch is removed and only the epistemic branch is trained.
> >
> > On the GLV2 misclassification detection task (Table 5), the fully joint model outperforms both single-branch variants across all metrics, indicating that each type of uncertainty estimation benefits from the presence of the other branch during training. In OOD detection (Table 6), the epistemic uncertainty from the joint model also achieves substantially higher AUC and AUPR than the epistemic-only variant (PAPILA AUC: 0.877–0.771), demonstrating that the joint formulation yields a more distribution-aware and robust representation.
> >
> > This improvement arises from the complementary objectives of the two branches. For aleatoric uncertainty, the prediction-consistency constraint used in the epistemic loss,  $ \min\_{\mathbf{m}'\_{l,n}} ||\hat{\mathbf{y}}'\_n - \hat{\mathbf{y}}\_n||\_1 $,  regularizes the shared feature extractor by discouraging perturbation-sensitive or unstable representations. This yields better-conditioned intermediate features for variance regression. Conversely, the aleatoric branch’s calibrated modeling of data-dependent variability provides an additional normalization signal to the backbone, helping the epistemic branch distinguish meaningful distributional deviations from sample-specific noise. Overall, these results confirm that CUPID’s two-branch design forms a synergistic training mechanism, with the joint model consistently producing more reliable and discriminative uncertainty estimates than either branch trained in isolation.
>
> **Table 5: Misclassification detection performance on GLV2 (Joint vs. separate branches).**
>
> | **Model**          |                   |   **Aleatoric**   |                   |                   |   **Epistemic**   |                   |
> | ------------------ | :---------------: | :---------------: | :---------------: | :---------------: | :---------------: | :---------------: |
> |                    |     **AUC ↑**     |    **AURC ↓**     |  **Spearman ↑**   |     **AUC ↑**     |    **AURC ↓**     |  **Spearman ↑**   |
> | **Joint**          | **0.870 ± 0.002** | **0.018 ± 0.001** | **0.941 ± 0.004** | **0.769 ± 0.015** | **0.034 ± 0.002** | **0.701 ± 0.051** |
> | **Alea. separate** |   0.863 ± 0.003   |   0.019 ± 0.001   |   0.899 ± 0.035   |         —         |         —         |         —         |
> | **Epis. separate** |         —         |         —         |         —         |   0.744 ± 0.017   |   0.043 ± 0.005   |   0.699 ± 0.014   |
>
> **Table 6: OOD detection performance (Joint vs. separate branches).**
>
> | **Model** | **Type** | **PAPILA** AUC ↑  | **PAPILA** AUPR ↑ | **ACRIMA** AUC ↑  | **ACRIMA** AUPR ↑ | **CIFAR-10** AUC ↑ | **CIFAR-10** AUPR ↑ |
> | --------- | -------- | ----------------- | ----------------- | ----------------- | ----------------- | ------------------ | ------------------- |
> | **Alea.** | Joint    | 0.379 ± 0.027     | 0.333 ± 0.007     | 0.717 ± 0.029     | 0.661 ± 0.027     | **0.983 ± 0.005**  | **0.998 ± 0.001**   |
> | **Alea.** | Separate | 0.508 ± 0.097     | 0.385 ± 0.052     | 0.739 ± 0.071     | 0.661 ± 0.066     | 0.969 ± 0.027      | 0.995 ± 0.005       |
> | **Epis.** | Joint    | **0.877 ± 0.032** | **0.854 ± 0.027** | **0.978 ± 0.010** | **0.984 ± 0.007** | 0.898 ± 0.054      | 0.991 ± 0.005       |
> | **Epis.** | Separate | 0.771 ± 0.051     | 0.707 ± 0.073     | 0.972 ± 0.010     | 0.978 ± 0.009     | 0.844 ± 0.049      | 0.986 ± 0.005       |

---

> ### Author Response · Authors · 2025-11-22
> **Reply to Q3**
>
> **4. Q3:** *The aleatoric branch directly regresses variance from intermediate feature activations. How is aleatoric uncertainty trained in classification without ground-truth noise labels? Are the variances calibrated?*
>
> **R4:** Although classification does not provide ground-truth noise levels, CUPID estimates aleatoric uncertainty by adopting a heteroscedastic Brier-style likelihood, which follows the same probabilistic principle as heteroscedastic regression [5], but is defined on the probability simplex. Specifically, we treat the Softmax output $\hat{\mathbf{y}}\_n'$ as a continuous regression target and model the per-sample predictive noise through an input-dependent variance. This yields a valid heteroscedastic likelihood over class probabilities without requiring explicit noise annotations.
>
> Formally, the aleatoric branch predicts the log-variance
> $\mathbf{s}\_n = \log(\hat{\boldsymbol{\sigma}}\_n^2)$,
> and the corresponding negative log-likelihood becomes:
> $
> \mathcal{L}\_{\text{alea}}
> = \frac{1}{N} \sum\_{n=1}^{N}
> \left[
> \frac{1}{2}\exp(-\mathbf{s}\_n)
> \left|| \mathbf{y}\_n - \hat{\mathbf{y}}\_n' \right||\_2^2
> +
> \frac{1}{2}\mathbf{s}\_n
> \right],
> $
> where $\mathbf{y}\_n$ is the one-hot class label. This heteroscedastic Brier-style likelihood allows the model to learn sample-dependent noise by attenuating the loss for intrinsically ambiguous examples, while the $\frac{1}{2}\mathbf{s}\_n$ term prevents trivial inflation of variance. Since CUPID predicts $\mathbf{s}\_n$ (log-variance) rather than $\hat{\boldsymbol{\sigma}}\_n^2$ directly, no additional normalization or constraints are required for stable and calibrated training. Overall, this formulation provides a principled and likelihood-based way to train aleatoric uncertainty in classification settings, even in the absence of explicit noise supervision.
>
> Concerning the reviewer’s question on calibration, in the revised appendix, we report uncertainty calibration error (UCE) and the risk–uncertainty curve area (rAULC) [4] for the classification model, both of which are established metrics for measuring aleatoric calibration in classification. CUPID-Alea achieves consistently good UCE and rAULC compared to other methods, demonstrating that the predicted variances are indeed well calibrated.
>
> **Table S2: Calibration results on GLV2.**
>
> | **Method**        | **rAULC ↑**           | **UCE ↓**             |
> | ----------------- | --------------------- | --------------------- |
> | **CUPID Alea.**   | **0.840 ± 0.003**     | *0.042 ± 0.004* (2nd) |
> | **MC Dropout**    | 0.682 ± 0.009         | 0.052 ± 0.019         |
> | **Rate-in**       | *0.762 ± 0.011* (2nd) | **0.038 ± 0.009**     |
> | **IGRUE**         | 0.375 ± 0.024         | 0.145 ± 0.012         |
> | **PostNet Alea.** | 0.419 ± 0.019         | 0.166 ± 0.031         |
> | **PostNet Epis.** | 0.184 ± 0.066         | 0.254 ± 0.013         |
> | **BNN**           | 0.747 ± 0.017         | 0.050 ± 0.007         |
> | **DEC**           | –0.627 ± 0.056        | 0.417 ± 0.039         |
>
> We thank the reviewer again for the valuable comments. We have clarified that CUPID is jointly trained by default, added explicit unified-model results, strengthened the theoretical explanation distinguishing CUPID from reconstruction-based approaches, and expanded calibration/ablation analyses. We hope these additions address all concerns and highlight CUPID’s novelty as the first plug-in module enabling joint, interpretable estimation of aleatoric and epistemic uncertainty within frozen deep networks.
>
> **Reference**
>
> [1] An Jinwon and Cho Sungzoon. Variational Autoencoder Based Anomaly Detection Using Reconstruction Probability. Special Lecture on IE, 2(1): 1–18, 2015.
>
> [2] Lu Mi, Hao Wang, Yonglong Tian, Hao He, and Nir N. Shavit. Training-free uncertainty estimation for dense regression: Sensitivity as a surrogate. In Proceedings of the AAAI Conference on Artificial Intelligence, volume 36, pp. 10042–10050, 2022.
>
> [3] Tobias Riedlinger, Matthias Rottmann, Marius Schubert, and Hanno Gottschalk. Gradient-based quantification of epistemic uncertainty for deep object detectors. In Proceedings of the IEEE/CVF Winter Conference on Applications of Computer Vision, pp. 3921–3931, 2023.
>
> [4] Hanjing Wang and Qiang Ji. Epistemic uncertainty quantification for pre-trained neural networks. In Proceedings of the IEEE/CVF Conference on Computer Vision and Pattern Recognition, pp. 11052–11061, 2024.
>
> [5] Alex Kendall and Yarin Gal. What uncertainties do we need in bayesian deep learning for computer vision? In Advances in Neural Information Processing Systems, volume 30, pp. 5580–5590, 2017.

---

### Official Review · Reviewer_o9ug · 2025-11-01

**Soundness:** 1
**Presentation:** 2
**Contribution:** 1
**Rating:** 2
**Confidence:** 4

**Summary:**

This paper proposes a joint estimation of aleatoric and epistemic uncertainty throuhgh a plug-and-play  auxililary module insertable at any layer of a pretrained deep model. The aleatoric uncertainty is estimated through an uncertainty branch and the epistemic uncertainty is estimated through the difference between original and perturbed outputs. The aleatoric and epistemic uncertainty loss is applied together to train the auxiliary module. The usefulness of the quantified uncertainty has been demonstrated in medical image misclassification detection, OOD detection, and image resolutions.

**Strengths:**

**1. Well-written:** The paper is well-written and well-motivated with good literature review.

**2. Experimental results:** The paper is enriched with numerous experimental evaluations of their methods in different tasks and demonstrated their success with AUC, AURC, spearman and Pearson coefficients. They also evaluated the calibration error. Their proposed method perform better in almost all cases.

**Weaknesses:**

**1.Confusing interpretation of epistemic uncertainty:** The epistemic uncertainty has been considered as the discrepancy between the original and the perturbed prediction. However, why this discrepancy captures the model's lack of knowledge is not clear from the paper.

**2. Limited novelty in aleatoric uncertainty:** The estimation of aleatoric uncertainty seems similar with BayesCap [2] and therefore that part has limited novelty.


[1] Depeweg, S., Hernandez-Lobato, J.M., Doshi-Velez, F. and Udluft, S., 2018, July. Decomposition of uncertainty in Bayesian deep learning for efficient and risk-sensitive learning. In International conference on machine learning (pp. 1184-1193). PMLR

[2] Upadhyay, U., Karthik, S., Chen, Y., Mancini, M. and Akata, Z., 2022, October. Bayescap: Bayesian identity cap for calibrated uncertainty in frozen neural networks. In European Conference on Computer Vision (pp. 299-317). Cham: Springer Nature Switzerland.

**Questions:**

The epistemic uncertainty part of the loss function seems similar with the identity mapping of the BayesCap function. Is it possible to draw a direct comparison with this paper's loss with BayesCap?

---

> ### Author Response · Authors · 2025-11-22
> **Reply to W1**
>
> We thank the reviewer for the constructive feedback. We appreciate the recognition of our paper’s clear writing, motivation, and extensive experimental evaluation. Below, we address each concern point-by-point and clarify the novelty, intuition, and theoretical grounding of CUPID.
>
> **1. W1:**  *Confusing interpretation of epistemic uncertainty: The epistemic uncertainty has been considered as the discrepancy between the original and the perturbed prediction. However, why this discrepancy captures the model's lack of knowledge is not clear from the paper.*
>
> **R1:** We thank the reviewer for highlighting the need to clarify the intuition behind our epistemic uncertainty measure. We improve the explanation as follows.
>
> CUPID estimates epistemic uncertainty by measuring the output discrepancy induced by a **learned, prediction-preserving perturbation**:
> $$ U\_{\text{epis}}(\mathbf{x}) := ||F\_l(\mathbf{m}\_{l,n}) - F\_l(\mathbf{m}'\_{l,n})||\_1.$$
> This formulation is grounded in the classical view that epistemic uncertainty reflects how unstable or underdetermined the model is in a neighborhood of the input representation [1]. Rather than injecting random noise or relying on multiple forward passes, CUPID learns a structured perturbation direction that maximizes representational deviation while keeping the prediction nearly invariant.
>
> **Why this corresponds to epistemic uncertainty.** A first-order Taylor expansion around $\mathbf{m}\_{l,n}$ yields:
>
> $U\_{\text{epis}}(\mathbf{x})
> \approx
> \underbrace{
> ||
> \nabla\_{\mathbf{m}\_{l,n}} F\_l(\mathbf{m}\_{l,n})
> \cdot
> (\mathbf{m}'\_{l,n}-\mathbf{m}\_{l,n})
> ||\_1
> }\_{%
> \text{Sensitivity}
> \times
> \text{Deviation}
> }$
>
> Here, $\mathbf{m}\_{l,n}$ denotes the intermediate feature at layer $l$ for sample $n$,
> $\mathbf{m}'\_{l,n}$ is its perturbed counterpart, and $F\_l(\cdot)$ is the prediction head operating on layer-$l$ features. This decomposition aligns CUPID with established epistemic-uncertainty theory:
>
> (1) **Sensitivity.** The Jacobian magnitude $\nabla F\_l$ reflects how fragile the model is to perturbations, a well-known indicator of epistemic uncertainty in Bayesian, ensemble, and gradient-based approximations [2, 3].
>
> (2) **Deviation.** The learned deviation $||\mathbf{m}'\_{l,n}-\mathbf{m}\_{l,n}||\_1$ quantifies how far the representation can move while keeping the prediction unchanged.
>
> Thus, CUPID captures two complementary aspects: the model’s local instability (model uncertainty) and the degree to which the representation can wander off the learned data manifold (distributional uncertainty). In-distribution misclassified samples often exhibit high sensitivity, while OOD samples induce abnormally large deviation. CUPID therefore provides a unified estimate of epistemic uncertainty that responds to both failure modes.
>
> This interpretation is further validated experimentally: incorporating the differential feature loss ($\max\_{\mathbf{m}'\_{l,n}} ||\mathbf{m}'\_{l,n}-\mathbf{m}\_{l,n}||\_1$) substantially improves CUPID's epistemic performance under distribution shift (PAPILA AUC increases from $0.839$ to $0.877$; AUPR from $0.790$ to $0.854$). These results confirm that the structured perturbation and its induced discrepancy faithfully capture epistemic uncertainty. We added the Table 4 in revised paper section 4.4.
>
> **Table 4: Performance of differential feature loss on OOD task. "No max" means remove $-||\mathbf{m}\_{l,n} - \mathbf{m}\_{l,n}'||\_1$ in the loss function. Best-performing results for each metric are highlighted in bold.**
>
> | **Method** | **Type** | **PAPILA AUC ↑**  | **PAPILA AUPR ↑** | **ACRIMA AUC ↑**  | **ACRIMA AUPR ↑** | **CIFAR-10 AUC ↑** | **CIFAR-10 AUPR ↑** |
> | ---------- | -------- | ----------------- | ----------------- | ----------------- | ----------------- | ------------------ | ------------------- |
> | Max        | Alea.    | 0.379 ± 0.027     | 0.333 ± 0.007     | 0.717 ± 0.029     | 0.661 ± 0.027     | 0.983 ± 0.005      | 0.998 ± 0.001       |
> | No max     | Alea.    | 0.389 ± 0.026     | 0.338 ± 0.009     | 0.739 ± 0.042     | 0.696 ± 0.055     | **0.988 ± 0.003**  | **0.999 ± 0.000**   |
> | Max        | Epis.    | **0.877 ± 0.032** | **0.854 ± 0.027** | **0.978 ± 0.010** | **0.984 ± 0.007** | 0.898 ± 0.054      | 0.991 ± 0.005       |
> | No max     | Epis.    | 0.839 ± 0.017     | 0.790 ± 0.054     | 0.977 ± 0.006     | 0.982 ± 0.005     | 0.875 ± 0.024      | 0.989 ± 0.002       |

---

> ### Author Response · Authors · 2025-11-22
> **Reply to W2（1）**
>
> **2. W2:** *The estimation of aleatoric uncertainty seems similar with BayesCap [2] and therefore that part has limited novelty.*
>
> **R2:** We appreciate the reviewer’s concern and are happy to clarify why CUPID’s aleatoric component is distinct from BayesCap.
>
> **CUPID novelty** The novelty of CUPID’s aleatoric component within the overall framework. CUPID is, to our knowledge, the **first model-preserving plug-in module that jointly estimates aleatoric and epistemic uncertainty from intermediate layers of any frozen network**. Prior methods either (1) require retraining or architectural changes (BNN, EDL), (2) estimate only epistemic uncertainty (RUE, gradient-based methods), or (3) estimate only aleatoric uncertainty (standard heteroscedastic regression). CUPID is the only approach that enables:
>
> * joint estimation of both uncertainty types in a **single** auxiliary module,
> * **insertion at arbitrary intermediate layers** of any pretrained network, enabling layer-wise uncertainty analysis,
> * single-pass inference without ensembling or repeated sampling.
>
> While the variance regression branch uses a classical negative log-likelihood, its contribution is unique because it is trained jointly with the reconstruction branch, and the **two branches reinforce each other during optimization**. In the revised manuscript, we added an ablation study in section 4.4 shows that joint training improves both aleatoric and epistemic performance, highlighting the benefit of CUPID’s unified design.
>
> > **Joint vs. Separate Training.**  Both the reconstruction branch (epistemic) and the uncertainty branch (aleatoric) in CUPID are present and jointly optimized. To evaluate whether this two-branch architecture provides mutual benefit, we conduct an ablation study in which we remove one branch entirely and train the remaining branch in isolation. Specifically, (1) *Alea. separate* denotes a model where the epistemic branch is removed and only the aleatoric branch is trained, and (2) *Epis. separate* denotes a model where the aleatoric branch is removed and only the epistemic branch is trained.
> >
> > On the GLV2 misclassification detection task (Table 5), the fully joint model outperforms both single-branch variants across all metrics, indicating that each type of uncertainty estimation benefits from the presence of the other branch during training. In OOD detection (Table 6), the epistemic uncertainty from the joint model also achieves substantially higher AUC and AUPR than the epistemic-only variant (PAPILA AUC: 0.877–0.771), demonstrating that the joint formulation yields a more distribution-aware and robust representation.
> >
> > This improvement arises from the complementary objectives of the two branches. For aleatoric uncertainty, the prediction-consistency constraint used in the epistemic loss,  $ \min\_{\mathbf{m}'\_{l,n}} ||\hat{\mathbf{y}}'\_n - \hat{\mathbf{y}}\_n||\_1 $,  regularizes the shared feature extractor by discouraging perturbation-sensitive or unstable representations. This yields better-conditioned intermediate features for variance regression. Conversely, the aleatoric branch’s calibrated modeling of data-dependent variability provides an additional normalization signal to the backbone, helping the epistemic branch distinguish meaningful distributional deviations from sample-specific noise. Overall, these results confirm that CUPID’s two-branch design forms a synergistic training mechanism, with the joint model consistently producing more reliable and discriminative uncertainty estimates than either branch trained in isolation.
>
> **Table 5: Misclassification detection performance on GLV2 (Joint vs. separate branches).**
>
> | **Model**          |                   |   **Aleatoric**   |                   |                   |   **Epistemic**   |                   |
> | ------------------ | :---------------: | :---------------: | :---------------: | :---------------: | :---------------: | :---------------: |
> |                    |     **AUC ↑**     |    **AURC ↓**     |  **Spearman ↑**   |     **AUC ↑**     |    **AURC ↓**     |  **Spearman ↑**   |
> | **Joint**          | **0.870 ± 0.002** | **0.018 ± 0.001** | **0.941 ± 0.004** | **0.769 ± 0.015** | **0.034 ± 0.002** | **0.701 ± 0.051** |
> | **Alea. separate** |   0.863 ± 0.003   |   0.019 ± 0.001   |   0.899 ± 0.035   |         —         |         —         |         —         |
> | **Epis. separate** |         —         |         —         |         —         |   0.744 ± 0.017   |   0.043 ± 0.005   |   0.699 ± 0.014   |

---

> ### Author Response · Authors · 2025-11-22
> **Reply to W2（2）**
>
> **Table 6: OOD detection performance (Joint vs. separate branches).**
>
> | **Model** |          | **PAPILA** AUC ↑  | **PAPILA** AUPR ↑ | **ACRIMA** AUC ↑  | **ACRIMA** AUPR ↑ | **CIFAR-10** AUC ↑ | **CIFAR-10** AUPR ↑ |
> | --------- | -------- | ----------------- | ----------------- | ----------------- | ----------------- | ------------------ | ------------------- |
> | **Alea.** | Joint    | 0.379 ± 0.027     | 0.333 ± 0.007     | 0.717 ± 0.029     | 0.661 ± 0.027     | **0.983 ± 0.005**  | **0.998 ± 0.001**   |
> | **Alea.** | Separate | 0.508 ± 0.097     | 0.385 ± 0.052     | 0.739 ± 0.071     | 0.661 ± 0.066     | 0.969 ± 0.027      | 0.995 ± 0.005       |
> | **Epis.** | Joint    | **0.877 ± 0.032** | **0.854 ± 0.027** | **0.978 ± 0.010** | **0.984 ± 0.007** | 0.898 ± 0.054      | 0.991 ± 0.005       |
> | **Epis.** | Separate | 0.771 ± 0.051     | 0.707 ± 0.073     | 0.972 ± 0.010     | 0.978 ± 0.009     | 0.844 ± 0.049      | 0.986 ± 0.005       |
>
> **Difference with BayesCap** BayesCap is really an interesting work. Although both methods employ a likelihood-based variance term, the underlying objectives, design choices, and applicability differ substantially: BayesCap models aleatoric uncertainty by training a Bayesian autoencoder directly on the output image space of a pretrained SR/denoising model, which makes it task-specific and reconstruction-driven. Our CUPID module instead operates in intermediate feature space, jointly modeling aleatoric and epistemic uncertainty through structured, prediction-preserving perturbations, enabling applicability across classification and regression tasks.
> The key conceptual and empirical distinctions are summarized in Table S1 below. We also have already compared the performance of BayesCap and CUPID in the manuscript section 4.3. CUPID Aleatoric achieves superior performance across all natural image datasets (Pearson $>$ 0.52, AUSE $<$ 0.12, UCE $<$ 0.05) while BayesCap got (Pearson $>$ 0.42, AUSE $<$ 0.12, UCE $<$ 0.10).
>
> **Table S1: Comparison between BayesCap and CUPID.**
>
> | **Aspect**                 | **BayesCap**                                                 | **CUPID (Ours)**                                             |
> | -------------------------- | ------------------------------------------------------------ | ------------------------------------------------------------ |
> | **Operating space**        | Operates on the **output image space** of a pretrained model (“Cap”), limiting it to image-to-image tasks (SR, denoising). | Operates on **intermediate feature space**, enabling use across classification, detection, and regression tasks. |
> | **Uncertainty types**      | Models only **aleatoric** uncertainty.                       | Jointly models **aleatoric + epistemic** uncertainty with shared gradients that benefit both. |
> | **Modeling philosophy**    | Uses a Bayesian autoencoder to **reconstruct the output distribution**; reconstruction error is used to improve aleatoric uncertainty estimation. | Uses **prediction-invariant feature perturbations**, not reconstruction. The perturbation mechanism naturally aligns with epistemic uncertainty (see R1). |
> | **Performance (Sec. 4.3)** | Pearson > 0.42, AUSE < 0.12, UCE < 0.10 on natural-image datasets. | Pearson > 0.52, AUSE < 0.12, UCE < 0.05 (best overall calibration and ranking). |

---

> ### Author Response · Authors · 2025-11-22
> **Reply to Q1**
>
> **3. Q1:** *The epistemic uncertainty part of the loss function seems similar with the identity mapping of the BayesCap function. Is it possible to draw a direct comparison with this paper's loss with BayesCap?*
>
> **R3.** We thank the reviewer for the opportunity to clarify this point. Below we present the two loss functions and provide a direct comparison using unified notation. This makes the conceptual differences explicit.
>
> **CUPID loss:**
>
> $\mathcal{L}\_{\text{cupid}} =
> \frac{1}{N} \sum\_{n=1}^{N}
> [
> \frac{1}{2} \exp(-\mathbf{s}\_n)
> ||\mathbf{y}\_n - \hat{\mathbf{y}}\_n'||\_2^2  + \frac{1}{2}\mathbf{s}\_n
> ]
> +
> \lambda\_2(
> \frac{1}{N} \sum\_{n=1}^{N}
> \big[
> ||\hat{\mathbf{y}}\_n - \hat{\mathbf{y}}\_n'||\_1 - \lambda\_1 ||\mathbf{m}\_{l,n}' - \mathbf{m}\_{l,n}||\_1
> \big]
> ).$
>
> Here, $\hat{\mathbf{y}}\_n$ is the output of the frozen base model, $\hat{\mathbf{y}}\_n'$ the output under CUPID's learned feature perturbation, and $\mathbf{y}\_n$ the ground truth.
>
> **BayesCap loss [4]:**
>
> $
> \mathcal{L}\_{\text{bayescap}} =
> \frac{1}{N}
> \sum\_{n=1}^{N}
> [
> \lambda ||\hat{\mathbf{y}}\_n - \tilde{\mathbf{y}}\_n||\_2^{2}+
> (
> \frac{||\tilde{\mathbf{y}}\_n - \mathbf{y}\_n||\_2}{\tilde{\alpha}\_n}
> )^{\tilde{\beta}\_n}-
> \log\frac{\tilde{\beta}\_n}{\tilde{\alpha}\_n}+
> \log\Gamma(\frac{1}{\tilde{\beta}\_n})
> ],
> $
>
> where $\tilde{\mathbf{y}}\_n$ is the reconstruction produced by the BayesCap uncertainty head, and $(\tilde{\alpha}\_n, \tilde{\beta}\_n)$ are the parameters of a generalized Gaussian likelihood.
>
> Although both methods contain an identity-mapping term, their objectives, modeling assumptions, and functional roles}differ fundamentally:
>
> (1) **Different likelihood families.** BayesCap uses a generalized Gaussian distribution, which is harder to optimize, requires learning both scale and shape parameters.  CUPID intentionally adopts the standard heteroscedastic Gaussian, which is numerically stable, broadly applicable, and compatible with both regression and classification.
>
> (2) **Different meaning of the identity-mapping term.** In BayesCap,  $||\hat{\mathbf{y}}\_n - \tilde{\mathbf{y}}\_n||\_2^2$ enforces that the BayesCap autoencoder reconstructs the same output as the pretrained model.  It is a regularizer used solely to stabilize aleatoric uncertainty estimation. In CUPID, the term $||\hat{\mathbf{y}}\_n - \hat{\mathbf{y}}\_n'||\_1$ serves the opposite purpose: it constrains the prediction to remain stable while allowing the feature representation to move as far as possible. Combined with the feature-deviation term $-\lambda\_1||\mathbf{m}\_{l,n}' - \mathbf{m}\_{l,n}||\_1$, it defines a maximal-perturbation search that is specifically designed to capture epistemic uncertainty. Thus the mathematical direction of the objective is fundamentally reversed.
>
> (3) **Different architectural principles.** BayesCap operates only on output space and reconstructs images. CUPID is an intermediate-layer plug-in that perturbs hidden features, supports arbitrary insertion depth, and enables layer-level epistemic analysis—something BayesCap cannot provide.
>
> (4) **Different uncertainty goals.** BayesCap models only aleatoric uncertainty. CUPID jointly models epistemic + aleatoric uncertainty in a single auxiliary module, with the two branches benefiting from shared optimization.
>
> We sincerely thank the reviewer once again for the thoughtful and constructive feedback. The clarifications provided above, together with the additional analyses and new ablation experiments, substantially strengthen the presentation and further reinforce the conceptual and empirical contributions of our work. We hope these responses satisfactorily address all concerns.
>
> **Reference**
>
> [1] Lu Mi, Hao Wang, Yonglong Tian, Hao He, and Nir N. Shavit. Training-free uncertainty estimation for dense regression: Sensitivity as a surrogate. In Proceedings of the AAAI Conference on Artificial Intelligence, volume 36, pp. 10042–10050, 2022.
>
> [2] Tobias Riedlinger, Matthias Rottmann, Marius Schubert, and Hanno Gottschalk. Gradient-based quantification of epistemic uncertainty for deep object detectors. In Proceedings of the IEEE/CVF Winter Conference on Applications of Computer Vision, pp. 3921–3931, 2023.
>
> [3] Hanjing Wang and Qiang Ji. Epistemic uncertainty quantification for pre-trained neural networks. In Proceedings of the IEEE/CVF Conference on Computer Vision and Pattern Recognition, pp. 11052–11061, 2024.
>
> [4] Uddeshya Upadhyay, Shyamgopal Karthik, Yanbei Chen, Massimiliano Mancini, and Zeynep Akata. Bayescap: Bayesian identity cap for calibrated uncertainty in frozen neural networks. In Proceedings of the European Conference on Computer Vision, pp. 299–317, 2022.

---

### Official Review · Reviewer_Lc84 · 2025-11-02

**Soundness:** 2
**Presentation:** 3
**Contribution:** 2
**Rating:** 4
**Confidence:** 4

**Summary:**

The paper proposes a plug-in module that estimates both aleatoric and epistemic uncertainty without retraining or changing the base network. The proposed CUPID is attached to intermediate layers of an existing network and includes two branches: an uncertainty branch that learns heteroscedastic variance for aleatoric uncertainty and a reconstruction branch that generates feature perturbations to quantify epistemic uncertainty. Experiments cover misclassification detection, out-of-distribution detection, and image super-resolution regression

**Strengths:**

+ CUPID generally outperforms or matches baselines like MC Dropout, Rate-in, PostNet, BNN, DEC, and BayesCap, with its two branches complementing each other across tasks.
+ A theoretical section provides a first-order Taylor expansion showing that epistemic uncertainty scales with the product of network sensitivity and feature deviation.

**Weaknesses:**

- The AU branch resembles standard heteroscedastic regression, and EU relies on output-preserving perturbations, which is close in spirit to prior work that estimates distributional shift via reconstruction error [RUE (Wang et al., 2023)]. It is unclear what the most significant novelty of this paper is.
- CUPID adds a learned module plus an extra forward of the perturbed path, but there is no computational cost analysis.
- The authors claim CUPID produces reliable uncertainty estimates, however, they do not check if high predicted uncertainty actually corresponds to higher error or misclassification rate. They only reports a calibration measure UCE in the regression (super-resolution) tasks, but not in classification or OOD detection.

**Questions:**

Can the authors explain why CUPID Aleatoric performs poorly on the PAPILA OOD detection experiment?

---

> ### Author Response · Authors · 2025-11-22
> **Reply to W1(1)**
>
> We sincerely thank the reviewer for the constructive and detailed feedback. We appreciate the reviewer’s positive assessment of our empirical performance, theoretical insight, and broad applicability. Below, we address each concern point-by-point and provide additional clarifications and experiments that reinforce the contribution and validity of CUPID.
>
> **1. W1:** *The AU branch resembles standard heteroscedastic regression, and EU relies on output-preserving perturbations, which is close in spirit to prior work that estimates distributional shift via reconstruction error [RUE (Wang et al., 2023)]. It is unclear what the most significant novelty of this paper is.*
>
> **R1.** We appreciate the reviewer’s question regarding novelty. CUPID is, to our knowledge, the **first model-preserving plug-in module that jointly estimates aleatoric and epistemic uncertainty from intermediate layers of any frozen network**. Prior approaches typically (1) require retraining or architectural modification (BNN, EDL), (2) capture only epistemic uncertainty (RUE, gradient methods), or (3) capture only aleatoric uncertainty (heteroscedastic regression) [1]. CUPID uniquely enables joint decomposition, layer-wise insertion, and requires only two forward passes without changing the base model.
>
> **Aleatoric branch.** While the variance regression follows standard heteroscedastic modeling [2], its role in CUPID is fundamentally different: **it contributes to a unified plug-in framework that simultaneously estimates both uncertainty types.** We have added an ablation study in section 4.4 of the paper showing that the jointly trained branches outperform separate training, highlighting the benefit of the unified design.
>
> > **Joint vs. Separate Training.**  Both the reconstruction branch (epistemic) and the uncertainty branch (aleatoric) in CUPID are present and jointly optimized. To evaluate whether this two-branch architecture provides mutual benefit, we conduct an ablation study in which we remove one branch entirely and train the remaining branch in isolation. Specifically, (1) *Alea. separate* denotes a model where the epistemic branch is removed and only the aleatoric branch is trained, and (2) *Epis. separate* denotes a model where the aleatoric branch is removed and only the epistemic branch is trained.
> >
> > On the GLV2 misclassification detection task (Table 5), the fully joint model outperforms both single-branch variants across all metrics, indicating that each type of uncertainty estimation benefits from the presence of the other branch during training. In OOD detection (Table 6), the epistemic uncertainty from the joint model also achieves substantially higher AUC and AUPR than the epistemic-only variant (PAPILA AUC: 0.877–0.771), demonstrating that the joint formulation yields a more distribution-aware and robust representation.
> >
> > This improvement arises from the complementary objectives of the two branches. For aleatoric uncertainty, the prediction-consistency constraint used in the epistemic loss,  $ \min\_{\mathbf{m}'\_{l,n}} ||\hat{\mathbf{y}}'\_n - \hat{\mathbf{y}}\_n||\_1 $,  regularizes the shared feature extractor by discouraging perturbation-sensitive or unstable representations. This yields better-conditioned intermediate features for variance regression. Conversely, the aleatoric branch’s calibrated modeling of data-dependent variability provides an additional normalization signal to the backbone, helping the epistemic branch distinguish meaningful distributional deviations from sample-specific noise. Overall, these results confirm that CUPID’s two-branch design forms a synergistic training mechanism, with the joint model consistently producing more reliable and discriminative uncertainty estimates than either branch trained in isolation.

---

> ### Author Response · Authors · 2025-11-22
> **Reply to W1(2)**
>
> **Table 5: Misclassification detection performance on GLV2 (Joint vs. separate branches).**
>
> | **Model**          |                   |   **Aleatoric**   |                   |                   |   **Epistemic**   |                   |
> | ------------------ | :---------------: | :---------------: | :---------------: | :---------------: | :---------------: | :---------------: |
> |                    |     **AUC ↑**     |    **AURC ↓**     |  **Spearman ↑**   |     **AUC ↑**     |    **AURC ↓**     |  **Spearman ↑**   |
> | **Joint**          | **0.870 ± 0.002** | **0.018 ± 0.001** | **0.941 ± 0.004** | **0.769 ± 0.015** | **0.034 ± 0.002** | **0.701 ± 0.051** |
> | **Alea. separate** |   0.863 ± 0.003   |   0.019 ± 0.001   |   0.899 ± 0.035   |         —         |         —         |         —         |
> | **Epis. separate** |         —         |         —         |         —         |   0.744 ± 0.017   |   0.043 ± 0.005   |   0.699 ± 0.014   |
>
> **Table 6: OOD detection performance (Joint vs. separate branches).**
>
> | **Model** | **Type** | **PAPILA** AUC ↑  | **PAPILA** AUPR ↑ | **ACRIMA** AUC ↑  | **ACRIMA** AUPR ↑ | **CIFAR-10** AUC ↑ | **CIFAR-10** AUPR ↑ |
> | --------- | -------- | ----------------- | ----------------- | ----------------- | ----------------- | ------------------ | ------------------- |
> | **Alea.** | Joint    | 0.379 ± 0.027     | 0.333 ± 0.007     | 0.717 ± 0.029     | 0.661 ± 0.027     | **0.983 ± 0.005**  | **0.998 ± 0.001**   |
> | **Alea.** | Separate | 0.508 ± 0.097     | 0.385 ± 0.052     | 0.739 ± 0.071     | 0.661 ± 0.066     | 0.969 ± 0.027      | 0.995 ± 0.005       |
> | **Epis.** | Joint    | **0.877 ± 0.032** | **0.854 ± 0.027** | **0.978 ± 0.010** | **0.984 ± 0.007** | 0.898 ± 0.054      | 0.991 ± 0.005       |
> | **Epis.** | Separate | 0.771 ± 0.051     | 0.707 ± 0.073     | 0.972 ± 0.010     | 0.978 ± 0.009     | 0.844 ± 0.049      | 0.986 ± 0.005       |
>
> **Epistemic branch vs. RUE.** CUPID is conceptually distinct from RUE. RUE-type methods **reconstruct the input** from intermediate features and use  **reconstruction error** as a proxy for distributional shift. This inherently restricts them to relatively shallow or mid-level layers: when features come from deeper layers, the discrepancy between the input image and the deep representation dominates the reconstruction error, making it difficult to disentangle uncertainty from inevitable information loss. Importantly, however, the **final layers** are the most **task-specific** and carry the most relevant information for detecting misclassification. By not leveraging these deep representations, **RUE-based models [3, 4] underperform on misclassification detection. As we shown in the manuscript Table 2, IGRUE obtains only 0.642 AUC on GLV2, whereas CUPID Aleatoric achieves 0.870**.
>
> CUPID instead introduces a **feature-level, output-preserving perturbation**: it learns a deviation $m'\_l - m\_l$ that maximally alters intermediate features while keeping $F\_l(m'\_l)$ close to $F\_l(m\_l)$. Thus, CUPID explicitly models the relationship between intermediate representations and the final prediction, capturing sensitivity in the deepest and most task-relevant layers, a capability that RUE cannot provide. Figure 2 has been revised to make this distinction clearer.
>
> In summary, CUPID's novelty lies not in individual loss components but in its **model-preserving, layer-adaptive, and jointly trained** architecture, which offers a new and principled route to unified uncertainty estimation.

---

> ### Author Response · Authors · 2025-11-22
> **Reply to W2**
>
> **2. W2:** *CUPID adds a learned module plus an extra forward of the perturbed path, but there is no computational cost analysis.*
>
> **R2.**  Thank you for pointing this out. We have added a detailed runtime comparison in the revised appendix as Table 27 and Table 28. CUPID introduces only a lightweight plug-in module and requires exactly **two forward passes** (original and perturbed), without retraining or modifying the base model. Consequently, its inference cost is substantially lower than methods that rely on repeated stochastic sampling (MC Dropout, in-noise) or multiple full model evaluations (PostNet, BNN). Empirically, CUPID achieves competitive or lower testing time while maintaining strong performance.
>
> **Table 27: Runtime comparison between training and testing phases for uncertainty estimation in classification.**
>
> | **Operation**  | **Training Time (s)** | **Testing Time (s)** |
> | -------------- | --------------------- | -------------------- |
> | **CUPID**      | 4776.38               | 10.06                |
> | **MC Dropout** | /                     | 49.57                |
> | **Rate-in**    | 2.85                  | 48.46                |
> | **IGRUE**      | 5292.41               | 11.26                |
> | **PostNet**    | 49932.08              | 51.04                |
> | **BNN**        | 54974.84              | 217.06               |
> | **DEC**        | 1615.58               | 6.46                 |
>
> **Table 28: Runtime comparison between training and testing phases for uncertainty estimation in regression.**
>
> | **Operation**   | **Training Time (s)** | **Testing Time (s)** |
> | --------------- | --------------------- | -------------------- |
> | **CUPID**       | 26511.74              | 1.02                 |
> | **BayesCap**    | 81158.45              | 0.79                 |
> | **in-rotate**   | /                     | 2.03                 |
> | **in-noise**    | /                     | 2.86                 |
> | **med-dropout** | /                     | 2.28                 |
> | **med-noise**   | /                     | 2.47                 |

---

> ### Author Response · Authors · 2025-11-22
> **Reply to W3**
>
> **3. W3:** *The authors claim CUPID produces reliable uncertainty estimates, however, they do not check if high predicted uncertainty actually corresponds to higher error or misclassification rate. They only reports a calibration measure UCE in the regression (super-resolution) tasks, but not in classification or OOD detection.*
>
> **R3.** We thank the reviewer for highlighting this point. In the revised appendix, we now report two established reliability metrics for classification: **the uncertainty calibration error (UCE) and the risk-uncertainty curve area (rAULC)** [5]. These metrics directly quantify whether samples with higher predicted uncertainty exhibit higher empirical error.
>
> For OOD detection, the standard UCE is not directly applicable because misclassification error cannot be defined for OOD samples whose labels differ fundamentally from the in-distribution taxonomy (e.g., GLV2 vs. CIFAR-10). To address this, we follow recent practice and evaluate OOD-UCE, which measures the alignment between predicted uncertainty and the empirical probability of being out-of-distribution:
>
> $\mathrm{OOD\text{-}UCE}=
> \sum\_{b} \frac{|B\_b|}{N} | \frac{1}{|B\_b|} \sum\_{i\in B\_b} o\_i - \frac{1}{|B\_b|}\sum_{i\in B_b} \tilde{u}_i|,
> $
>
> where $o\_i$ is the OOD indicator and $\tilde{u}\_i$ is the normalized uncertainty.
>
> The full tables are included in the updated appendix as below. These added results confirm that (1) CUPID’s uncertainty positively correlates with classification error (highly ranked by rAULC and UCE), and (2) CUPID exhibits strongalignment between uncertainty and OOD likelihood.
>
> >  Table 16 summarizes uncertainty calibration performance for both misclassification and OOD settings. CUPID consistently ranks first or second across all metrics, demonstrating strong uncertainty–error correlation and stable calibration. On GLV2, CUPID Aleatoric achieves the highest rAULC (0.840), while CUPID Epistemic obtains the best UCE (0.033), indicating excellent uncertainty ranking and calibration. For OOD detection, CUPID achieves the lowest OOD-UCE on PAPILA (0.115) and competitive results on ACRIMA (0.226 and 0.240). Notably, DEC attains the best OOD-UCE on ACRIMA (0.198), but this comes at the cost of severely degraded misclassification calibration (rAULC = –0.627, UCE = 0.417), suggesting that DEC improves OOD calibration only by sacrificing its ability to reflect in-distribution prediction errors. This contrast highlights CUPID’s balanced and reliable uncertainty modeling across both ID and OOD scenarios.
>
> **Table: Uncertainty calibration results for misclassification (GLV2) and OOD detection (PAPILA, ACRIMA, CIFAR-10).**
>
> | **Method**        | **GLV2** rAULC ↑      | **GLV2** UCE ↓        | **PAPILA** OOD-UCE ↓  | **ACRIMA** OOD-UCE ↓  | **CIFAR-10** OOD-UCE ↓ |
> | ----------------- | --------------------- | --------------------- | --------------------- | --------------------- | ---------------------- |
> | **CUPID Alea.**   | **0.840 ± 0.003**     | 0.042 ± 0.004         | 0.308 ± 0.026         | *0.226 ± 0.079* (2nd) | *0.273 ± 0.004* (2nd)  |
> | **CUPID Epis.**   | 0.649 ± 0.031         | **0.033 ± 0.008**     | **0.115 ± 0.016**     | 0.240 ± 0.006         | 0.633 ± 0.017          |
> | **MC Dropout**    | 0.682 ± 0.009         | 0.052 ± 0.019         | 0.257 ± 0.001         | 0.307 ± 0.045         | 0.699 ± 0.021          |
> | **Rate-in**       | *0.762 ± 0.011* (2nd) | *0.038 ± 0.009* (2nd) | 0.309 ± 0.009         | 0.411 ± 0.008         | 0.826 ± 0.019          |
> | **IGRUE**         | 0.375 ± 0.024         | 0.145 ± 0.012         | *0.154 ± 0.046* (2nd) | 0.252 ± 0.016         | 0.596 ± 0.010          |
> | **PostNet Alea.** | 0.419 ± 0.019         | 0.166 ± 0.031         | 0.179 ± 0.025         | 0.285 ± 0.036         | 0.500 ± 0.125          |
> | **PostNet Epis.** | 0.184 ± 0.066         | 0.254 ± 0.013         | 0.182 ± 0.104         | 0.289 ± 0.089         | 0.726 ± 0.082          |
> | **BNN**           | 0.747 ± 0.017         | 0.050 ± 0.007         | 0.268 ± 0.010         | 0.332 ± 0.012         | 0.705 ± 0.008          |
> | **DEC**           | –0.627 ± 0.056        | 0.417 ± 0.039         | 0.256 ± 0.027         | **0.198 ± 0.010**     | **0.248 ± 0.008**      |

---

> ### Author Response · Authors · 2025-11-22
> **Reply to W4**
>
> **4. P1:** *Can the authors explain why CUPID Aleatoric performs poorly on the PAPILA OOD detection experiment?.*
>
> **R4.** The weaker performance of CUPID Aleatoric on PAPILA is consistent with the underlying data characteristics. PAPILA is a **near-OOD** dataset: although collected from a different source, its images share similar anatomical structures and overall appearance with GLV2. The resulting domain shift is subtle (color tone, contrast) rather than noise-driven. Since aleatoric uncertainty reflects **data-dependent noise** rather than **distributional mismatch**, it is not expected to increase strongly under such mild shifts. We provide visual samples between ID dataset GLV2 and OOD datasets in the revised appendix (Fig. 7) to illustrate this behavior.
>
>
> We thank the reviewer again for the valuable comments. We believe these clarifications and new experiments strengthen the paper and address all concerns.
>
> **Reference**
>
> [1] Ke Zou, Zhihao Chen, Xuedong Yuan, Xiaojing Shen, Meng Wang, and Huazhu Fu. A review of uncertainty estimation and its application in medical imaging. Meta-Radiology, pp. 100003, 2023.
>
> [2] Xuanlong Yu, Gianni Franchi, Jindong Gu, and Emanuel Aldea. Discretization-induced dirichlet posterior for robust uncertainty quantification on regression. In Proceedings of the AAAI Conference on Artificial Intelligence, volume 38, pp. 6835–6843, 2024.
>
> [3] Li Rong Wang, Thomas C. Henderson, and Xiuyi Fan. An uncertainty estimation model for algorithmic trading agent. In Proceedings of the International Conference on Intelligent Autonomous Systems, pp. 459–465, 2023.
>
> [4] Lennard Korte, Li Rong Wang, and Xiuyi Fan. Confidence estimation in analyzing intravascular optical coherence tomography images with deep neural networks. In Proceedings of the IEEE Conference on Artificial Intelligence (CAI), pp. 358–364, 2024.
>
> [5]Hanjing Wang and Qiang Ji. Epistemic uncertainty quantification for pre-trained neural networks. In Proceedings of the IEEE/CVF Conference on Computer Vision and Pattern Recognition, pp. 11052–11061, 2024.

---

### Author Response · Authors · 2025-12-02

Dear Area Chair,

Given the disruption caused by the OpenReview incident and the halted discussion period, we provide a concise summary of how our rebuttal addressed the core reviewer concerns. None of the reviewers could respond before the platform failure, so we hope this assists your evaluation.

**1. Contributions and Reasons for Consideration**

- CUPID is, to our knowledge, the **first plug-in module that jointly estimates aleatoric and epistemic uncertainty from intermediate layers of any frozen network**.
- Strong empirical performance across **misclassification**, **OOD detection**, and **regression**, consistently outperforming baselines.
- Ability to insert CUPID at multiple depths enables analysis of **uncertainty propagation** through the network, offering new diagnostic insight.

**2. Reviewer-Identified Strengths**

Reviewers consistently highlighted several strengths:

* **Comprehensive experiments** (Lc84, o9ug, 7Juj): Strong empirical foundation across diverse tasks, baselines, and metrics.
* **Novel and practical plug-in design** (D4Tj, 7Juj): Applies to pretrained models without retraining, with low overhead and strong real-world applicability.
* **Clear theoretical grounding** (Lc84): The sensitivity × deviation formulation was viewed as principled and well motivated.
* **Clarity of presentation** (o9ug, 7Juj): Reviewers found the paper well written, well organized, and well motivated.

**3. Resolution of Core Reviewer Concerns**

Reviewers converged on several questions that we addressed thoroughly.

**(1) CUPID vs. Reconstruction-Based Methods (RUE by Lc84 and autoencoder by D4Tj)**  Reviewers asked whether CUPID resembles reconstruction-based uncertainty estimation.  We clarified that:

- Prior methods minimize reconstruction error and treat residuals as uncertainty, which captures distributional shift but poorly reflects epistemic uncertainty and misclassification risk.
- CUPID does **not** minimize reconstruction error. Instead, CUPID optimizes **structured feature perturbations under prediction invariance**, yielding a sensitivity–deviation decomposition of epistemic uncertainty.
- Updated Fig. 2 and a toy example (Fig. 1) illustrate this mechanism. The appendix provide additional evidence demonstrating the relationship between perturbation/sensitivity and uncertainty.

Thus CUPID is a perturbation/sensitivity method, conceptually distinct from reconstruction-based approaches.

**(2) CUPID vs. Heteroscedastic Regression (BayesCap o9ug)**  We clarified that:

- Although CUPID uses a heteroscedastic likelihood, the variance branch plays a different role: it provides **aleatoric** estimates while enabling a **unified plug-in architecture** where epistemic and aleatoric uncertainties are jointly optimized. Our added ablation (Sec. 4.4) shows that joint training outperforms separate branches, also directly addressing D4Tj’s question about whether the components benefit each other.
- BayesCap models **only aleatoric** uncertainty and is tied to **output-space reconstruction** for SR/denoising.
- CUPID operates in feature space and applies to **any predictive model**. Direct comparisons show CUPID Aleatoric achieves higher Pearson and lower UCE/AUSE across datasets.

**(3) Additional Experiments** We added:

- Comprehensive calibration analyses where CUPID ranks **first or second** across metrics.
- Training/inference runtime comparisons showing **competitive overhead**.
- A λ₁ sensitivity study confirming CUPID’s **robustness** across tasks.
- Improved dataset visualizations clarifying **AU–EU differences** between GLV2 and HAM10000.
- OOD ablation experiments demonstrating that the **differential feature loss** further improves performance.

**4. Understanding Score Divergence**

The spread (8–4–4–2) appears to stem from differences in clarity rather than methodological issues.

- **7Juj (8)** found the method well motivated with strong empirical support.
- **Lc84 and D4Tj (4)** mainly sought clarification on distinctions between CUPID and reconstruction-based or heteroscedastic methods. These points were fully addressed through improved figures, clearer notation, and additional experiments.
- **o9ug (2)** provided a brief review focused on comparisons with specific methods, while acknowledging the good motivation and sufficient experiments. Our rebuttal resolved these comparison-related concerns with detailed contrasts in mechanism and performance.

Across all reviews, none questioned the validity of the method or experimental results; the issues centered on conceptual clarity, which we have fully addressed.

Given the resolved concerns and the interrupted discussion period, we respectfully request that the paper be evaluated based on the clarified evidence rather than the initial numerical average. We believe the contributions are meaningful and of direct relevance to the ICLR community.

Thank you for your time and consideration.

Sincerely,

The authors

---

### Meta-Review · Area_Chair_UD7J · 2026-01-07

**Summary:**

The main contribution of this paper is a joint aleatoric and epistemic uncertainty estimation approach. The reviewers have concerns about the novelty, to which the authors responded that their estimation is done jointly, while the existing approaches either estimate the two types of uncertainties separately or have other limitations such as requiring architectural changes. With the joint estimation, this paper's proposed approach reduces the computational cost compared to the related work while maintaining a good accuracy. This is verified by the added running time comparisons.

**Reviewer Concerns:**

Based on my reading, it seems the reviewers concerns have been well addressed. However, I'm not very familiar with this field and cannot say it for certain. The authors have done a good rebuttal in general, with detailed responses and new results.

**Reviewer Scores:**

I believe some reviewers would have raised their scores.

---

### Decision · Program_Chairs · 2026-01-26

Accept (Poster)